# Glutamate Supplementation Regulates Nitrogen Metabolism in the Colon and Liver of Weaned Rats Fed a Low-Protein Diet

**DOI:** 10.3390/nu17091465

**Published:** 2025-04-26

**Authors:** Da Jiang, Jing Zhang, Yun Ji, Zhaolai Dai, Ying Yang, Zhenlong Wu

**Affiliations:** State Key Laboratory of Animal Nutrition and Feed Science, China Agricultural University, Beijing 100193, China; jayd1994@163.com (D.J.); s20233040807@cau.edu.cn (J.Z.); jean500@163.com (Y.J.); daizhaolai@cau.edu.cn (Z.D.); cauvet@163.com (Y.Y.)

**Keywords:** glutamate, low-protein, mTOR, nitrogen metabolism, rats

## Abstract

**Background**: Glutamate, a nutritionally non-essential amino acid, is a key intermediate in nitrogen metabolism. Despite more studies on its functional role in intestine health, it remains unknown how glutamate regulates nitrogen metabolism in animals fed a low-protein diet. **Methods**: Herein, we investigated the effects of glutamate supplementation on colonic amino acid transport, barrier protein expression, microbiota alterations, fecal nitrogen emissions, hepatic amino acid transport, and protein synthesis in weaned rats. **Results**: We found that protein restriction diminished the mucus thickness, reduced goblet cell numbers, and the expression of *EAAT3*, *y^+^LAT2* in the colon. In contrast, glutamate supplementation reversed these effects, increasing the colon length and enhancing the expression of ZO-1, Occludin, and Claudin-1 in the colon. At the genus level, glutamate increased the abundance of *Lactococcus* and *Clostridia_sensu_stricto_18*. Additionally, glutamate supplementation resulted in an increased apparent nitrogen digestibility, reduced the ratio of fecal nitrogen to total nitrogen intake, and increased the ratio of fecal microbial nitrogen to total nitrogen intake. Protein restriction decreased the mRNA level of *ATP1A1*, *EAAT3*, *SNAT9/2*, and *ASCT2*, and the protein level of p-mTOR, mTOR, p-mTOR/mTOR, and p-p70S6K/p70S6K as well as p-4EBP1/4EBP1 in the liver. These effects were reversed by glutamate supplementation. **Conclusions**: In conclusion, glutamate supplementation upregulates amino acid transporters and barrier protein expression in the colon, modulates microbiota composition to reduce fecal nitrogen excretion, and enhances amino acid transport and protein synthesis in the liver by activating the mTOR/p70S6K/4EBP1 pathway, which influences nitrogen metabolism in weaned rats fed a low-protein diet.

## 1. Introduction

Protein–energy malnutrition (PEM) is a condition characterized by inadequate nutrition due to insufficient intake or increased requirements of protein and/or energy [1]. The global incidence of PEM has decreased in recent years owing to advancements in medical care, public health initiatives, and the expansion of the food industry [2]. According to a 2021 report by the United Nations Children’s Fund, in 2020, 149.2 million children under the age of five were stunted, and another 45.4 million exhibited wasting [3]. Outside of Africa, the number of stunted children has been declining across all regions. More than half of the children affected by wasting reside in South Asia, while Asia as a whole accounts for over three-quarters of the world’s severely wasting children. At the national level, significant progress has been made in reducing stunting, with nearly two-thirds of countries achieving at least partial success in meeting their targets. However, it continues to pose a significant health challenge across all age groups, particularly for children [4,5]. Although the global prevalence of PEM in children is declining [2], regional disparities persist; for example, the number of cases is increasing in developing countries, particularly in South Asia and sub-Saharan Africa [6,7]. In 2019, it was estimated that 21.3% (approximately 144 million) of children under the age of five globally were stunted, with 36% concentrated in sub-Saharan Africa and South Asia [8]. From 2012 to 2019, the prevalence of stunting in sub-Saharan Africa decreased from 34.5% to 31.1%, yet remained below the global target level [8]. However, when measured in absolute numbers rather than proportions, sub-Saharan Africa is the only sub-region where the number of stunted children has increased in recent years [8]. The occurrence of child stunting is influenced by a range of complex factors, including maternal nutritional status, health and education levels, pregnancy intervals, child birth weight, vaccination coverage, infection rates, infant feeding practices, household economic conditions, food security, and environmental factors [9,10]. However, there is limited literature on whether glutamate can alleviate malnutrition in infants or children after weaning due to inadequate protein intake.

Dietary proteins and amino acids are primarily hydrolyzed, absorbed, and metabolized in the small intestine before reaching the colon. In the colon, amino acids are transported into colonic epithelial cells via amino acid transporters on the cell membrane and play a critical role in the synthesis of various proteins, including tight junction proteins and mucins [11]. Additionally, changes in dietary protein levels significantly influence the amount of protein and amino acids entering the large intestine, thereby altering the number of bacteria utilizing nitrogen as a substrate [12,13] and further modifying the composition of the colonic microbiota. Key bacteria involved in amino acid metabolism include *Bacteroides*, *Propionibacterium*, *Streptococcus*, *Actinomyces*, *Lactococcus*, *Fusobacterium*, and *Clostridium* [14]. The metabolites produced by these bacteria encompass not only short-chain fatty acids and branched-chain fatty acids [15], but also other compounds such as ammonia, phenols, and indole [16]. Ultimately, undigested dietary proteins, nucleic acids, and proteins derived from bacteria and intestinal shed cells are excreted in feces as sources of fecal nitrogen [17]. Notably, glutamate, as a functional amino acid, serves not only as the primary energy source for intestinal epithelial cells [18], but also plays essential roles in nutrient absorption and transport, protein synthesis [19], signal transduction, and barrier function maintenance [20,21,22].

The gut is a site of significant proteolytic activity, predominantly mediated by the microbiota. Within the gut microbiota, bacterial species ranging from sugar-fermenting bacteria to obligate amino acid fermenters exhibit the capacity to metabolize peptides and amino acids. *Bacteroides* and *Clostridium* species dominate aromatic amino acid metabolism, converting tryptophan into indole-3-acetic acid and skatole, while *Streptococcus* species are associated with phenyl derivative production [23,24,25]. *Clostridium*, *Enterobacteriaceae*, and *Desulfovibrio* species metabolize sulfur-containing amino acids through cysteine desulfhydrase or cystathionine enzymes, producing hydrogen sulfide, which plays dual roles in mucosal damage or protection [26,27]. Amine-generating bacteria, including *Clostridium* and *Bacteroides*, mediate the decarboxylation of tryptophan to serotonin and polyamines (e.g., spermidine), influencing gut–brain axis signaling and epithelial function [25,28,29]. Furthermore, *Bacteroides*, *Clostridium*, and *Klebsiella* species contribute to ammonia production via urea hydrolysis or amino acid deamination, with excessive ammonia impairing colonocyte metabolism [14]. Collectively, these bacterial activities regulate colonic nitrogen balance and influence host health outcomes.

The liver is a critical organ for nitrogen nutrient turnover and the primary site of amino acid metabolism in the body. Amino acids are released into the portal vein circulation from the intestine [11,30] and are transported to hepatocytes, where they participate in catabolic redistribution (26–42%) and protein synthesis [31]. In the liver, amino acids undergo deamination to produce ammonia, which is then converted into urea via the urea cycle [32]. Most urea (over 80%) is excreted in urine by the kidneys through blood transport, while approximately 10% enters the intestine via the bile duct and is eventually excreted in feces [33]. Amino acids serve not only as essential nutrient substrates but also play a critical role in dynamic regulation through the complex liver signaling network. For instance, branched-chain amino acids (e.g., leucine) regulate protein degradation in the autophagy–lysosome pathway via the Sestrin2-GATOR2 axis [34]. Meanwhile, glutamine maintains the balance between gluconeogenesis and lipid oxidation by activating AMPK phosphorylation [35]. Notably, the regulatory functions of amino acids are tightly coupled with their metabolic states. Recent studies have demonstrated that under conditions of sufficient amino acids, SIRT4 inhibits the activity of ornithine transcarbamylase through deamination modification at the K307 site, thereby limiting ammonia production. Conversely, during amino acid deficiency, the general control nonderepressible 2 (GCN2)-eukaryotic initiation factor 2α (eIF2α)-activating transcription factor 4 (ATF4) axis is activated to upregulate SIRT4 expression, enhancing urea cycle flux and alleviating hepatic encephalopathy [36]. In this regulatory network, glutamate acts as a central hub [37]. Specifically, glutamate generates α-ketoglutarate through transamination, driving the tricarboxylic acid cycle. Additionally, glutamate combines with cysteine via glutamate–cysteine ligase to synthesize glutathione, a major antioxidant that protects liver cells from oxidative damage [38,39].

In the present study, we investigated the effects of glutamate supplementation on nitrogen metabolism in the colon and liver of weaned rats fed a low-protein diet, aiming to understand the regulatory effect of amino acids on protein metabolism.

## 2. Materials and Methods

### 2.1. Animals and Diets

The animal handling procedures complied with established ethical standards and received approval from the Institutional Animal Care and Use Committee at China Agricultural University (AW70114202-1-05). In total, 27 male Sprague Dawley rats, aged 3 weeks, purchased from BEIJING HFK BIOSCIENCE Co., Ltd. (Beijing, China), were individually housed and fed the American Institute of Nutrition (AIN)-93G diet, formulated in accordance with the guidelines set forth by the American Institute of Nutrition. The animals were allowed ad libitum access to food for a 7-day acclimatization period. Ambient conditions maintained a temperature of 24 °C with a 12 h light–dark cycle, while the rats had unrestricted access to drinking water throughout the experimental period [40].

Following the acclimatization period, randomization and blinding were employed to allocate the weaned rats (28 days of age) to one of three treatment groups: a normal crude protein diet (NCP) group, a low-crude-protein diet supplemented with alanine (LCP + Ala), or a low-crude-protein diet supplemented with 2.07% glutamate (LCP + Glu). In total, 27 healthy rats were randomly assigned to three treatment groups, with 9 rats in each group. Each rat constituted an independent experimental unit. The experimental period in this study was set to 18 days. This duration was chosen because rats weaned at 28 days of age enter a rapid growth phase over the subsequent 2–3 weeks (during which their weight doubles), making them particularly sensitive to nutritional interventions [41]. Additionally, previous studies have utilized a time window of 14–21 days to assess the effects of low-protein diets [42,43,44,45], and the design of this study aligns with these durations. Furthermore, by day 18 of the experiment, a highly significant difference in body weight was observed between the LCP + Ala group and the LCP + Glu group. Considering both statistical validity and ethical concerns (minimizing animal use time), the experimental period was ultimately determined to be 18 days. In our study, the NCP diet strictly adhered to the AIN-93G standard formulation. The primary amino acid sources were casein (comprising 20%) and cystine (comprising 0.3%). Since casein alone provides adequate essential amino acids (e.g., lysine and methionine), and the AIN-93G guidelines do not require supplementation with synthetic amino acids other than cystine, no additional individual amino acids were added to the NCP group. Equal amounts of the 10 essential amino acids (EAAs) were added to both the LCP + Ala diet and the LCP + Glu diet, ensuring consistent nutrient levels of EAAs across all three groups. Additionally, the glutamate content was 4.12% in the NCP group and 2.06% in the LCP + Ala group. After adding 2.07% glutamate (purity 99.5%), the glutamate content in the LCP + Glu group diet increased to 4.12%, aligning with the dietary glutamate content of the NCP group. Meanwhile, alanine was selected as it is a neutral amino acid suitable for regulating iso-nitrogen levels in the diet. This ensured the iso-nitrogen characteristics of both the LCP + Ala and LCP + Glu groups, thereby excluding any potential interference from nitrogen content differences and allowing us to focus on the functional specificity of glutamate (rather than its nitrogen contribution). In addition, all diets were designed to be isocaloric, providing a consistent total energy content of 4.35 kcal/g across all three treatment groups. The feed was provided by Xiao Shu You Tai (Beijing) Biotechnology Co., Ltd. (Beijing, China), and detailed information on ingredients and nutrients is shown in Table 1 and Appendix A. Throughout the 18-day experimental period, all rats had ad libitum access to water and their respective diets, allowing for unrestricted consumption. From days 15 to 18 of the experiment, the rats were transitioned from group housing (three rats per cage) to individual housing. Feces from each rat were collected and weighed daily. Additionally, any signs of disease were documented as “adverse events”, and affected rats were immediately euthanized. Notably, no adverse events were observed throughout the study, and all rats remained healthy. Importantly, no data were excluded during the sample testing process, ensuring that each experimental group maintained nine experimental units for all data analyses.

### 2.2. Fecal Sample Collection and Pretreatment

Fecal samples were collected from each rat over a 72 h period on days 15 to 18 using metabolic cages. The samples were dried to constant weight at 40 °C in a constant-temperature incubator. Subsequently, the fecal samples were individually ground into powder using a mortar and pestle, and precisely weighed to 0.2 g before being placed in EP tubes for sealed storage and subsequent analysis [46].

### 2.3. Determination of Fecal Nitrogen and Fecal Microbial Nitrogen

Fecal nitrogen was measured using the Kjeldahl method (GB/T 6432-94) [47]. Fecal microbial nitrogen was determined as follows: 0.2 g of fecal samples was diluted with normal saline at a ratio of 1:5, mixed thoroughly, and centrifuged at 250× *g* for 15 min at 4 °C. The supernatant was then subjected to further centrifugation at 14,500× *g* for 30 min at 4 °C. After removing the supernatant, the precipitate, which contained microbial cells [48], was washed twice with 0.5 mL of normal saline (each wash involved mixing followed by centrifugation). Subsequently, 0.5 mL of cell lysis buffer (pH 7.2, containing 0.5% sodium dodecyl sulfate and 0.5% sodium deoxycholate) was added. After homogenization, the mixture was centrifuged again. A 0.1 mL aliquot of the supernatant was taken and mixed with 0.1 mL of display reagent. The microbial nitrogen content was determined using the Nanjing Jiancheng Coomassie Brilliant Blue method-total protein determination kit.

### 2.4. Quantitation of mRNA Expression by Quantitative Real-Time PCR

Total RNA of colon and liver was extracted using TRIZOL reagent in accordance with established protocols. The extracted RNA was subsequently reverse-transcribed into complementary DNA (cDNA) using the High-Capacity cDNA Archive Kit from Applied Biosystems, following the manufacturer’s guidelines meticulously [49]. The resulting cDNA served as the template for polymerase chain reaction (PCR) analysis, targeting genes associated with energy metabolism, amino acid transporters, taste receptors, and metabolic amino acid receptors, employing standardized procedures throughout. For quantitative PCR amplification, primer sequences specific to the target genes were utilized, as detailed in Appendix A. These primers were carefully designed to ensure both specificity and efficiency in gene amplification.

### 2.5. Western Blot Analysis

Frozen colon and liver tissues were subjected to homogenization and lysis in ice-cold radioimmunoprecipitation assay (RIPA) lysis buffer, consisting of 50 mmol/L Tris-HCl (pH 7.4), 150 mmol/L NaCl, 1% NP-40, 0.1% SDS, 1.0 mmol/L PMSF, 1.0 mmol/L Na_3_VO_4_, 1.0 mmol/L NaF, and a protease inhibitor cocktail obtained from Roche Applied Science (Rockford, IL, USA). The cell lysate was then centrifuged at 12,000× *g* for 15 min at 4 °C to remove cellular debris. Protein concentration was determined using the BCA protein assay kit from Applygen Technologies Inc. (Beijing, China). Equal amounts of protein were separated on either 8% or 12% SDS-PAGE gels and subsequently transferred onto PVDF membranes (Millipore; Billerica, MA, USA). The membranes underwent blocking with a solution of 5% BSA at room temperature for durations of either 60 min prior to incubation with appropriately diluted primary antibodies [50]. Antibodies specific to proteins such as p-mTOR, mTOR, p-p70S6K, p70S6K, p-4EBP1, 4EBP1, zonula occludens (ZO)-1, Occludin, and Claudin-1 were procured from Invitrogen (Carlsbad, CA, USA). Following this step, blots were stripped and re-probed with an anti-glyceraldehyde-3-phosphate dehydrogenase (GAPDH) antibody from Santa Cruz Biotechnology to verify equal loading across samples. After incubation with horseradish peroxidase-conjugated secondary antibodies, the chemiluminescence signal was visualized using the Super Enhanced Chemiluminescence Kit provided by Applygen Technologies Inc. (Beijing, China). The density of bands was quantified utilizing Image J 1.53k software. Meanwhile, representative blots are displayed in figures and data from nine independent biological replicates were analyzed for each group.

### 2.6. Mucin Staining

The colon tissue segments (1 cm) were fixed in 4% paraformaldehyde, subsequently embedded in paraffin, and sectioned into slices of 5 μm thickness. Mucins were visualized through staining with Alcian Blue reagent obtained from Vector Laboratories, Burlingame, CA, in accordance with the manufacturer’s instructions. Briefly, the tissue sections were incubated with acetic acid solution for 3 min before being immersed in Alcian Blue solution for 30 min. The sections were then briefly rinsed (10–30 s) in acetic acid solution to eliminate excess Alcian Blue. Finally, the sections underwent incubation with nuclear fast red solution for 5 min followed by a 5 min wash with water [51]. The stained sections were observed under a microscope (Zeiss, Jena, Germany).

### 2.7. H&E Staining

Colon morphology was evaluated through hematoxylin and eosin (H&E) staining. The assessment involved procedures such as dehydration, embedding, sectioning, and staining of tissue samples [52]. The mucus thickness was quantified using computer-assisted microscopy (Micrometrics TM; Nikon ECLIPSE E200, Tokyo, Japan).

### 2.8. DNA Extraction and 16S rRNA Sequencing

Fresh colonic contents, collected under aseptic conditions, were promptly stored at −80 °C until further analysis. Subsequent DNA extraction procedures were performed following the instructions provided by a DNA extraction kit obtained from Omega Bio-Tek, Winooski, VT, USA. Following a quality assessment via 1% agarose gel electrophoresis, the isolated DNA was subjected to PCR amplification using primers [338F (5′-ACT CCT ACG GGA GGC AGC AG-3′) and 806R (5′-GGA CTA CHVGGG TWT CTAAT-3′)] that are specific to the V3–V4 hypervariable region of 16S rDNA [53]. The sequencing of the purified PCR products was subsequently conducted in accordance with the specifications of the Illumina MiSeq platform (Illumina, San Diego, CA, USA). Operational Taxonomic Units (OTUs) were clustered at a similarity threshold of 97% using UPARSE methodology. Data analysis was executed utilizing the Majorbio Cloud Platform based in Shanghai, China.

### 2.9. Statistical Analysis

The data are presented as means ± standard error of the mean (SEM) and were analyzed using a one-way analysis of variance (ANOVA). Variations among the means were evaluated employing the Duncan multiple comparison method. Statistical analyses were performed with SPSS statistical software (SPSS for Windows, version 26.0, Chicago, IL, USA). Prior to the analysis of variance, all continuous variables were verified for normality by the Shapiro–Wilk test. The results show that the data distribution conforms to the hypothesis of normality (*p* > 0.05), thus satisfying the precondition of the parameter test. Data with non-normal distribution can usually be processed by data transformation or non-parametric test, but no such adjustment is required in this study. A significance level of *p* < 0.05 was established to indicate statistical significance, while *p* < 0.01 was deemed highly significant.

## 3. Results

### 3.1. Effects of Glutamate Supplementation on Colon Length, Colon Morphology and Mucin Abundance in Weaned Rats with Protein Restriction

After reducing the crude protein (CP) levels, no significant change was observed in colon length. However, supplementation with glutamate resulted in a significant increase in colon length (*p* < 0.01) (Figure 1C). Reducing the CP levels led to a significant decrease in both mucus thickness (Figure 1A,D) and goblet cell numbers (*p* < 0.01) (Figure 1B,E). Furthermore, glutamate supplementation significantly restored mucus thickness (*p* < 0.01) and increased goblet cell numbers (*p* < 0.01).

### 3.2. The Impacts of Glutamate Supplementation in Protein-Restricted Diets on the Abundance of Genes Associated with Na/K-ATPase, Amino Acid Transporter, and Glutamate Receptor in the Colon

The decrease in CP levels was associated with a significant reduction in the expression levels of *excitatory amino acid transporter 3* (*EAAT3*) and *y^+^LAT2* (*p* < 0.05) (Figure 2E,I). Glutamate supplementation significantly increased the expression levels of these genes (*p* < 0.05). Following the decrease in CP levels, the expression levels of *adenosine triphosphate (ATP)1A4*, *EAAT4*, *xCT*, *4F2hc*, *SNAT2*, *alanine-serine-cysteine transporter 2* (*ASCT2*), *metabotropic glutamate receptor 1* (*mGluR1*), and *taste receptor type 1 subunit 1* (*T1R1*) also decreased significantly (*p* < 0.05) (Figure 2C,F–H,K–N). Additionally, glutamate supplementation did not significantly alter the expression levels of these genes. Meanwhile, protein restriction did not alter the expression levels of CaSR (Figure 2O), whereas glutamate supplementation markedly increased CaSR expression levels (*p* < 0.01). In contrast, the expression levels of *ATP1A1*, *ATP1A2*, *EAAT1*, and *B^0^AT1* remained relatively stable after decreasing the CP levels (Figure 2A,B,D,J), and no significant changes were observed following glutamate supplementation.

### 3.3. The Effects of Glutamate Supplementation in Protein-Restricted Diets on the Levels of TJ Proteins in the Colon

The expression of ZO-1, Occludin, and Claudin-1 in the colon did not change significantly in rats fed a low-protein diet (Figure 3). Notably, glutamate supplementation significantly increased the expression of these proteins (*p* < 0.05).

### 3.4. The Impacts of Glutamate on the Composition of the Microbial Community in Colon Contents

The microbiota composition in the colon of the weaned rats was analyzed by the Illumina HiSeq sequencing system. After size filtering, quality control, and chimera checking, a total of 65,605 ± 415, 68,231 ± 415, and 66,413 ± 413 reads were observed in the NCP, LCP + Ala, and LCP + Glu groups, respectively. The OTUs were obtained at a sequence similarity level of 97%. The rarefaction (Appendix A), Shannon curves (Appendix A), and rank abundance curves (Appendix A) tended to draw near the saturation plateau, reflecting that the sequencing capability was adequate for the diversity of these communities to be captured.

Additionally, no remarkable differences were found in the community richness (chao1 and sobs index) (Figure 4A,D), and the same was found for the community diversity (Simpson’s index) (Figure 4C) and community coverage (coverage index) (Figure 4B) of the colon microbiota. A shift in the gut microbial community was highlighted by the principal component analysis (PCA) in response to glutamate supplementation (Figure 4E). Consistently, the principal coordinate analysis (PCoA) results presented that the colonic microbial composition of the weaned rats that were (LCP + Glu group) or were not (LCP + Ala group) supplemented with glutamate was obviously distinguishable from that of the control (Figure 4F).

Following the Illumina MiSeq sequencing analysis, a Venn diagram of the OTUs showed that there were 992, 1162, and 1031 OTUs in the NCP, LCP + Ala, and LCP + Glu groups, respectively (Figure 5A). The relative abundance of bacteria at the phylum level for all samples is presented in Figure 5B. The top three dominant phylum (>0.1% in at least in one of the three groups) were *Firmicutes* (95.50%), *Actinobacteriate* (1.83%), and *Bacteroidota* (1.63%). At the genus level, 19 dominant genera were identified (>1% in at least in one of the three groups). The top five dominant genera were *Romboutsia*, *Lactobacillus*, *Clostridium_sensu_stricto_1*, *Enterococcus*, and *Staphylococcus*, with an average percentage of 25.01%, 20.42%, 18.67%, 4.71%, and 3.34%, respectively (Figure 5C). The linear discriminant analysis (LDA) with effect size measurements (LEfSe) for bacterial communities between dietary groups are shown in Figure 5D. Relative to the NCP and LCP + Ala group, the rats of the LCP + Glu group had a higher abundance of *Enterococcus*, *Weissella*, *Lactococcus*, *Candidatus_Saccharimonas*, *Clostridium_sensu_stricto_18*, and *Sphingobium*.

In the colon, the Spearman correlation analysis identified significant positive correlations between *Clostridium_sensu_stricto_1* abundance and body weight as well as average daily gain (Figure 6A). Moreover, *Lactococcus* was significantly positively correlated with colon length and the expression of *y^+^LAT2*, Claudin-1, and Occludin. Additionally, the microbial composition was related to biosynthesis and utilization (Figure 6B).

### 3.5. Effects of Glutamate Supplementation on Fecal Nitrogen Metabolism of Weaned Rats Fed a Low-Protein Diet

Compared with the control, protein restriction significantly decreased the fecal nitrogen (*p* < 0.01) and glutamate supplementation further reduced the fecal nitrogen (*p* < 0.01) of the weaned rats (Table 2). Meanwhile, protein restriction did not alter the ratio of fecal nitrogen to nitrogen intake; however, glutamate supplementation significantly lowered the ratio (*p* < 0.01). Additionally, protein restriction had no significant effect on the microbial nitrogen, the ratio of microbial nitrogen to fecal nitrogen, or apparent nitrogen digestibility, whereas glutamate supplementation significantly improved these parameters (*p* < 0.01).

### 3.6. Effects of Glutamate Supplementation on the Abundance of Genes Involved in Na/K-ATPase and Amino Acid Transporters in the Liver

After reducing CP levels, the expression of *EAAT1*, *xCT*, and *y^+^LAT2* remained unchanged (Figure 7D,G,I). However, glutamate supplementation significantly increased the expression of these genes (*p* < 0.05). Following CP reduction, the expression of *ATP1A1*, *EAAT3*, *SNAT9*, *SNAT2*, and *ASCT2* were significantly decreased (*p* < 0.05) (Figure 7A,E,L,N,O), while glutamate supplementation led to a significant increase in their expression levels (*p* < 0.05). In addition, after reducing the CP levels, the expression of *ATP1A2*, *ATP1A4*, *EAAT4*, *4F2hc*, *amino acid transporter B^0,+^* (*ATB^0,+^*), *Cat-1*, and *B^0^AT1* remained relatively stable, with no significant changes observed following glutamate supplementation (Figure 7B,C,F,H,J,K,M).

### 3.7. Effects of Glutamate Supplementation in Protein-Restricted Diets on the Abundance of mTOR/p70S6K/4EBP1 Signaling Pathways in the Liver

After reducing CP levels, the expression of p-mTOR, mTOR, p-mTOR/mTOR, p-p70S6K/p70S6K, and p-4EBP1/4EBP1 were significantly decreased (*p* < 0.05) (Figure 8A–D,G,J). Conversely, glutamate supplementation resulted in a significant increase in these expression levels (*p* < 0.05). Additionally, the expression of p70S6K and 4EBP1 were significantly increased after CP reduction (*p* < 0.05) (Figure 8F,I), whereas glutamate supplementation resulted in a significant decrease in their expression levels (*p* < 0.05). Furthermore, the phosphorylation levels of p70S6K and 4EBP1 remained relatively stable following CP reduction, with no significant changes observed after glutamate supplementation (Figure 8E,H).

## 4. Discussion

In low- and middle-income developing countries, malnutrition in infants and children due to inadequate protein intake after weaning is still on the rise [6,7]. In our study, a weaned rat model fed a low-protein diet was used to simulate this condition. We found that glutamate supplementation could enhance nitrogen metabolism in the colon and liver, suggesting that glutamate supplementation might potentially alleviate malnutrition caused by insufficient protein intake in infants and children. These findings provide valuable dietary guidance for managing PEM in children.

The primary sites of amino acid metabolism are the intestine [54] and the liver [55]. Dietary proteins are hydrolyzed and absorbed in the small intestine, with unabsorbed amino acids reaching the colon. The colon demonstrates significant potential in nitrogen metabolism. While the liver serves as the primary site for nitrogen metabolism (such as the urea cycle), recent studies indicate that the colon participates in the nitrogen cycle through microbial metabolism and host–microbiota interactions [56,57]. Consequently, glutamate supplementation may enhance the colon’s capacity to capture and metabolize nitrogen, thereby effectively mitigating nitrogen loss associated with low-protein diets. Moreover, the morphology and function of the colon are crucial to nitrogen metabolism. Low-protein diets frequently result in gut microbiome dysbiosis, compromising gut barrier function and elevating health risks [58,59]. Studies have shown that colonic goblet cells contain a large number of mucin-rich mucus particles, which are secreted into the intestinal lumen and form a dense mucus barrier on the mucus surface [60,61]. Glutamate plays a crucial role in maintaining mucus barrier functions [62,63]. In our study, we observed that protein restriction resulted in significant thinning of the colonic mucus layer and a marked reduction in the number of goblet cells compared to the control group. However, glutamate supplementation restored colonic mucus thickness and the number of goblet cells to control levels. This suggests that adequate glutamate availability is essential for preserving the mucus barrier. These findings align with the existing literature, which shows that glutamate activates the vagus nerve via intestinal glutamate receptors, such as calcium-sensing receptors, transmitting signals from the intestine to the central nervous system, and thus regulates intestinal mucus secretion [62,63].

Beyond alterations in the mucus barrier, we identified the effects of glutamate supplementation on amino acid transport and tight junction protein expression in the colon. Amino acids are unable to freely permeate the colonic cell membrane and must be actively transported across the membrane by amino acid transporters driven by Na/K-ATPase, thereby ensuring efficient nutrient supply [64]. In our study, we found that glutamate reversed the downregulation of the mRNA levels of acidic amino acid transporter *EAAT3* and neutral amino acid transporter *y^+^LAT2* induced by protein restriction. This indicates that protein restriction inhibits the transport of glutamate and neutral amino acids, whereas glutamate supplementation upregulates the expression of these transporters in the colon, thereby enhancing their transport efficiency and optimizing their utilization. Additionally, protein restriction did not affect the protein level of colonic tight junction (TJ) proteins ZO-1, Occludin, and Claudin-1, but glutamate supplementation significantly upregulated their expression levels. This suggests that protein restriction has no significant effect on the TJ protein expression of colonic epithelial cells, which is consistent with previous studies [65]. However, glutamate supplementation further enhanced the expression of TJ proteins in the colon, a finding that aligns with previous studies [21,66]. In summary, this study concludes that the nitrogen transport capacity of colon epithelial cells plays a critical role in regulating nitrogen metabolism under nutritional stress [67,68,69]. Building on the finding that protein restriction significantly reduces the thickness of the colon mucosa and the expression level of mucin, further investigations reveal that protein restriction markedly impairs the amino acid transport function of the colon. However, glutamate supplementation significantly enhances the colon’s nitrogen uptake capacity, improves nitrogen supply to the colon, and promotes colon mucus synthesis through the regulation of central nervous system function. These findings suggest that a key bottleneck limiting colon nitrogen metabolism under low-protein conditions is substrate availability (i.e., amino acid transport). Moreover, glutamate supplementation can effectively improve nitrogen metabolism by promoting amino acid transport and enhancing both the colon mucus barrier and mechanical barrier functions.

Compared to other parts of the intestine, the colon exhibits a more neutral pH, larger volume, and longer material retention time, which collectively provide more favorable conditions for microbial proliferation [70]. Moreover, the colon microbiota is more directly associated with mucosal barrier function (as evidenced by physical contact between the microbiota and epithelial cells), whereas the cecal contents primarily reflect microbial activity during the rapid fermentation stage [71]. Additionally, the sampling of colonic contents is more stable and less susceptible to variations in chyme flow velocity, and its data exhibit greater comparability with findings from previous studies [72]. Based on these considerations, this study focused on colonic microbiota as the primary research subject. Amino acids that enter the colon serve as a nitrogen source for coliform bacteria, supporting their growth and metabolic activities. Nitrogen is an essential element for the growth of coliform bacteria, playing a key role in macromolecular biosynthesis and acting as a precursor to secondary metabolites [73]. Amino acids, as organic nitrogen sources, are transported from the extracellular environment into the cytoplasm via bacterial membrane transporters [74]. Coliform bacteria are closely involved in nitrogen metabolism processes, and any changes in the microbiome can significantly impact the entire system [75]. In our study, the abundances of *Enterococcus*, *Weissella*, *Lactococcus*, *Candidatus*, *Clostridium_sensu_stricto_18*, and *Sphingobium* increased in the glutamate supplementation group. Coliform bacteria metabolize amino acids in ways that are closely related to host health [76], with *Lactococcus* and *Clostridium* being particularly important in amino acid metabolism [56]. *Clostridium* converts amino acids into ketoacids or saturated fatty acids via the Stickland reaction in the amino acid catabolic pathway, ultimately fermenting them into short-chain fatty acids and polyamines [56,77].

Additionally, *Lactococcus* species, as beneficial microorganisms [78], enhance barrier functions [79], prevent pathogen invasion [80,81], and boost host immunity [24,82,83] by producing antibacterial metabolites and bacteriocins, and by lowering the intestinal pH [82,84]. These mechanisms contribute to maintaining host intestinal health. The Spearman correlation analysis in our study revealed that the abundance of *Clostridium* was significantly positively correlated with body weight and average daily gain, while the abundance of *Lactococcus* was significantly positively correlated with colon length, TJ protein expression, and amino acid transport. The PICRUSt analysis revealed that the composition of microbial communities was associated with various biological functions, including biosynthesis and metabolic utilization. Therefore, we speculate that glutamate supplementation improves intestinal health by regulating the structure of coliform bacteria, thereby modulating bacterial nitrogen metabolism, promoting biosynthesis, and enhancing metabolic utilization.

Amino acids can be utilized by coliform bacteria and excreted in feces. Due to the growing environmental impact of animal waste, public awareness of environmental issues in animal production has also increased [85]. After microbial transformation, nitrogen is released into the atmosphere as ammonia, nitrous oxide, and nitric oxide. Additionally, it contributes to the pollution of groundwater and surface water via the leaching and runoff of nitrates and other nitrogen compounds [86]. In our study, we found that protein restriction did not affect the ratio of fecal nitrogen to nitrogen intake, while glutamate supplementation significantly reduced fecal nitrogen emissions and lowered the ratio of fecal nitrogen to nitrogen intake. Additionally, previous studies have shown that endogenous urea nitrogen produced by amino acid metabolism could be degraded by intestinal microorganisms [87] and converted into fecal microbial nitrogen [88], thereby reducing ammonia emissions. In our study, protein restriction had no significant effect on fecal microbial nitrogen emissions or the ratio of fecal microbial nitrogen to fecal nitrogen. Conversely, glutamate supplementation significantly improved these indicators. We speculate that changes in the structure of coliform bacteria after glutamate supplementation, particularly the increase in *Lactococcus* and *Clostridium* abundance, may enhance non-protein nitrogen utilization and fecal microbial nitrogen synthesis in weaned rats. On the other hand, dietary protein digestibility is an important index reflecting the degree of decomposition and absorption of dietary protein in the digestive tract. It can be divided into apparent digestibility and true digestibility based on whether endogenous fecal metabolic nitrogen is considered. In our study, we used apparent nitrogen digestibility to evaluate the nutritional value of diets in each group. The results showed that protein restriction had no significant effect on the apparent nitrogen digestibility, but glutamate supplementation significantly increased the apparent nitrogen digestibility. In conclusion, glutamate supplementation resulted in an increased apparent nitrogen digestibility, reduced the ratio of fecal nitrogen to total nitrogen intake, and increased the ratio of fecal microbial nitrogen to total nitrogen intake, thereby improving the nitrogen metabolism.

After proteins are digested in the gut, amino acids are transported to hepatocytes via the portal vein, a process mediated by amino acid transporters. Amino acid transporters not only directly control cellular uptake of amino acids but also serve as promoters of nutritional signals to corresponding receptors, thereby regulating gene expression, initiating cell signal transduction, and influencing physiological processes such as cell proliferation [89,90]. In the liver, amino acids regulate energy metabolism and participate in various biological processes as key signaling molecules. For example, amino acids act as neurotransmitters and activate the mTORC1 signaling pathway, which regulates important physiological functions such as gene transcription and protein translation [91,92]. When mTORC1 is activated by amino acids [93], eukaryotic initiation factor 4EBP1 is phosphorylated, thereby relieving elF4E-mediated inhibition of protein synthesis. Additionally, p70S6 kinase phosphorylates and drives rRNA transcription [37], ultimately enhancing the protein synthesis process. In our study, glutamate supplementation significantly restored the expression of transporters such as *EAAT3* and *SNAT9* and activated the mTOR/p70S6K/4EBP1 pathway in the liver. Notably, *EAAT3* is a high-affinity glutamate transporter, and its increased expression can promote glutamate transport, thereby activating the mTORC1 signaling pathway [94]. However, whether *EAAT3* mediates glutamate regulation and activation of the mTORC1 signaling pathway has not been confirmed. Moreover, recent studies have highlighted the importance of the amino acid transporter *SNAT9* in mTORC1 activation [95,96,97]. Therefore, we speculate that glutamate supplementation promotes the expression of amino acid transporters *EAAT3* and *SNAT9*, leading to the activation of the mTOR signaling pathway and consequently enhancing nitrogen metabolism in the liver.

We found that glutamate supplementation upregulates the expression of amino acid transporters and TJ proteins in the colon, modulates the composition of the intestinal microbiota, reduces fecal nitrogen excretion, and promotes amino acid transport and protein synthesis in the liver by activating the mTOR/p70S6K/4EBP1 pathway. These effects collectively influence nitrogen metabolism in weaned rats fed a low-protein diet. Meanwhile, in addition to glutamate, amino acids such as aspartate, arginine, and glutamine may contribute to improved growth performance through the regulation of nitrogen metabolism and enhancement of immune function. Amino acids such as aspartate, arginine, and glutamine in low-protein diets may contribute to improved growth performance through the regulation of nitrogen metabolism and enhancement of immune function. Aspartate and glutamate are both acidic amino acids that participate in the urea cycle and the tricarboxylic acid (TCA) cycle, playing crucial roles in energy metabolism. Furthermore, aspartate can modulate energy and lipid metabolism by regulating immune function and optimizing the composition of gut microbiota, thereby potentially enhancing growth performance [98,99]. However, aspartate and glutamate exhibit metabolic competition, such as sharing transporters, which could potentially influence their supplementation efficacy [43]. Due to insufficient endogenous synthesis, arginine is considered an essential amino acid for piglets. Dietary supplementation with arginine has been reported to strengthen the immune response of weaned piglets [100,101]. Nevertheless, high doses of arginine may exacerbate intestinal inflammation by increasing nitric oxide production, necessitating careful dose optimization. Glutamine, classified as a conditionally essential amino acid, serves as a precursor to glutamate. It can improve the growth performance of weaned piglets on low-protein diets while enhancing serum physicochemical parameters and antioxidant capacity [13]. Despite its benefits, glutamine is prone to rapid degradation in vivo, raising questions about the effectiveness of exogenous supplementation. In summary, although this study demonstrated that glutamate significantly enhances the growth of rats on low-protein diets, amino acids like aspartate and arginine may also contribute synergistically by regulating nitrogen metabolism and promoting protein synthesis.

Future studies should systematically assess the effects of various amino acid interventions, such as incorporating low-protein diets supplemented with aspartate, arginine, or glutamine. Additionally, introducing a glutamate deprivation model (e.g., using glutamate transporter inhibitors) could help observe whether growth inhibition and nitrogen metabolism disturbances can be reversed by exogenous glutamate supplementation. Simultaneously, it is essential to explore cost-effective and safe combination strategies to further enhance nutritional intervention programs for undernourished populations.

In addition, given the potential changes in amino acid transport, glutamate metabolism, immune system function, and nervous system development, as well as the adverse effects associated with discontinuing glutamate supplementation [102,103], we recommend short-term and intermittent multiple administrations of glutamate supplementation to minimize the risks associated with long-term consumption. Furthermore, additional trials are warranted to evaluate the efficacy and safety of long-term glutamate supplementation, including its potential toxicity and the physiological impacts following cessation after prolonged use.

In our study, glutamate, as a primary intestinal oxidizing fuel, a key neurotransmitter in the intestine, and a critical intermediate in nitrogen metabolism within the gastrointestinal tract and liver of weaned animals, can modulate the intestinal microbiota and enhance amino acid transport and protein synthesis in these organs. This suggests that glutamate may hold therapeutic potential to ameliorate malnutrition in children with PEM by enhancing nitrogen metabolism. Specifically, according to the experimental results in rats, incorporating 2.07% glutamate into a low-protein diet significantly enhances the growth and development of post-weaning rats. In our study, weaned rats were supplemented with 2.07% glutamate (based on an average daily feed intake of 20 g, resulting in an actual glutamate intake of approximately 0.414 g/day). Using the body surface area dose conversion formula (average child body weight: 20 kg, rat body weight: 0.15 kg, conversion coefficient: 0.18), the estimated child dose was calculated as 0.414 g ÷ 0.18 ≈ 2.3 g/day. Additionally, by considering the daily protein deficit (approximately 10–15 g) and the proportion of glutamate in dietary protein (5–8%) among children in developing countries, a conservative recommendation range of 2–4 g/day was derived through a comprehensive analysis. Therefore, if this insight is extended to human children, particularly those in developing countries with insufficient protein intake, it is hypothesized that daily supplementation of approximately 2–4 g of glutamate may offer potential benefits in addressing malnutrition. Nevertheless, the safety and practical efficacy of such supplementation must be rigorously evaluated through additional human trials.

## 5. Conclusions

In conclusion, glutamate supplementation promotes amino acid transport and enhances mucus and mechanical barrier functions in the colon. Meanwhile, glutamate supplementation modulates the structure of coliform bacteria, reduces fecal nitrogen emissions, increases the proportion of fecal microbial nitrogen in fecal nitrogen, and enhances the apparent nitrogen digestibility. Additionally, glutamate supplementation promotes amino acid transport, leading to the activation of the mTOR/p70S6K/4EBP1 signaling pathway in the liver, which influences nitrogen metabolism in weaned rats fed a low-protein diet supplemented with 10 EAAs.

## Figures and Tables

**Figure 1 nutrients-17-01465-f001:**
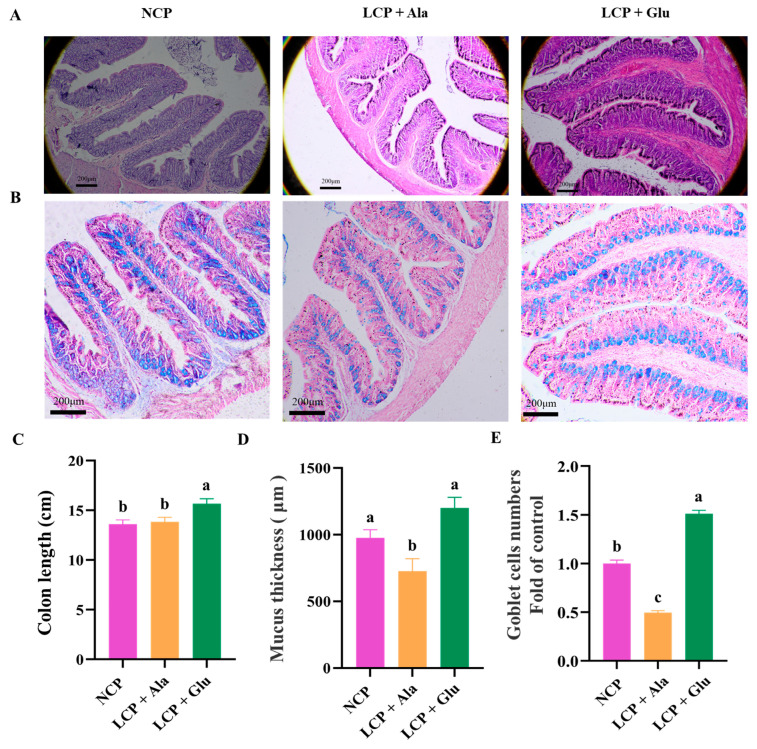
Colon length (**C**), H&E staining (**A**,**D**), and Alcian Blue staining (**B**,**E**) analysis in the colon tissue of 46-day-old weaned rats in the normal protein diet group, the low-protein diet group, and the low-protein diet supplemented with 2.07% glutamate group from days 28 through 46. NCP = normal crude protein diet, LCP + Ala = low-crude-protein diet supplemented with alanine, LCP + Glu = low-crude-protein diet supplemented with 2.07% glutamate. ^a–c^ Different superscripted letters in the same column indicate significant differences (*p* < 0.05).

**Figure 2 nutrients-17-01465-f002:**
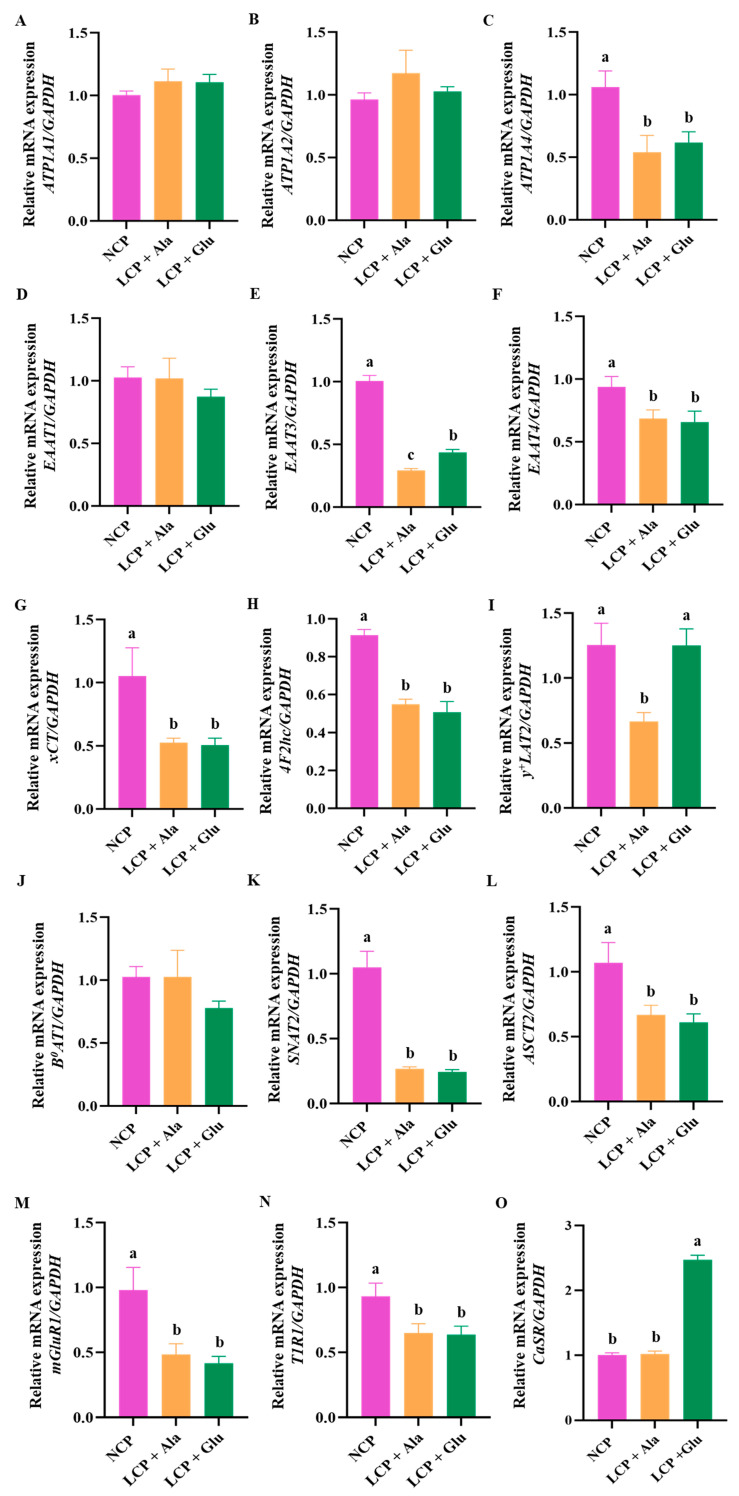
Real-time PCR analysis of Na/K-ATPase (**A**–**C**), amino acid transporter (**D**–**L**), and glutamate receptor gene (**M**–**O**) expression in the colon of 46-day-old weaned rats in the normal protein diet group, the low-protein diet group, and the low-protein diet supplemented with 2.07% glutamate group from days 28 through 46. NCP = normal crude protein diet; LCP + Ala = low-crude-protein diet supplemented with alanine; LCP + Glu = low-crude-protein diet supplemented with 2.07% glutamate. ^a–c^ Different superscripted letters in the same column indicate significant differences (*p* < 0.05).

**Figure 3 nutrients-17-01465-f003:**
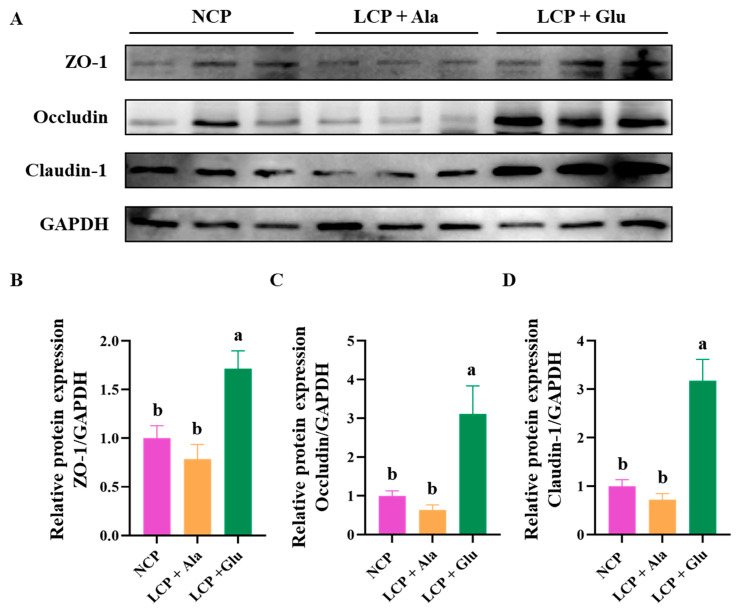
Western blot bands (**A**) and abundances analysis (**B**–**D**) of ZO-1, Occludin, and Claudin-1 proteins in the colon tissue of 46-day-old weaned rats in the normal protein diet group, the low-protein diet group, and the low-protein diet supplemented with 2.07% glutamate group from days 28 through 46. NCP = normal crude protein diet, LCP + Ala = low-crude-protein diet supplemented with alanine, LCP + Glu = low-crude-protein diet supplemented with 2.07% glutamate. Representative blots from n = 9 biological replicates are shown. ^a,b^ Different superscripted letters in the same column indicate significant differences (*p* < 0.05).

**Figure 4 nutrients-17-01465-f004:**
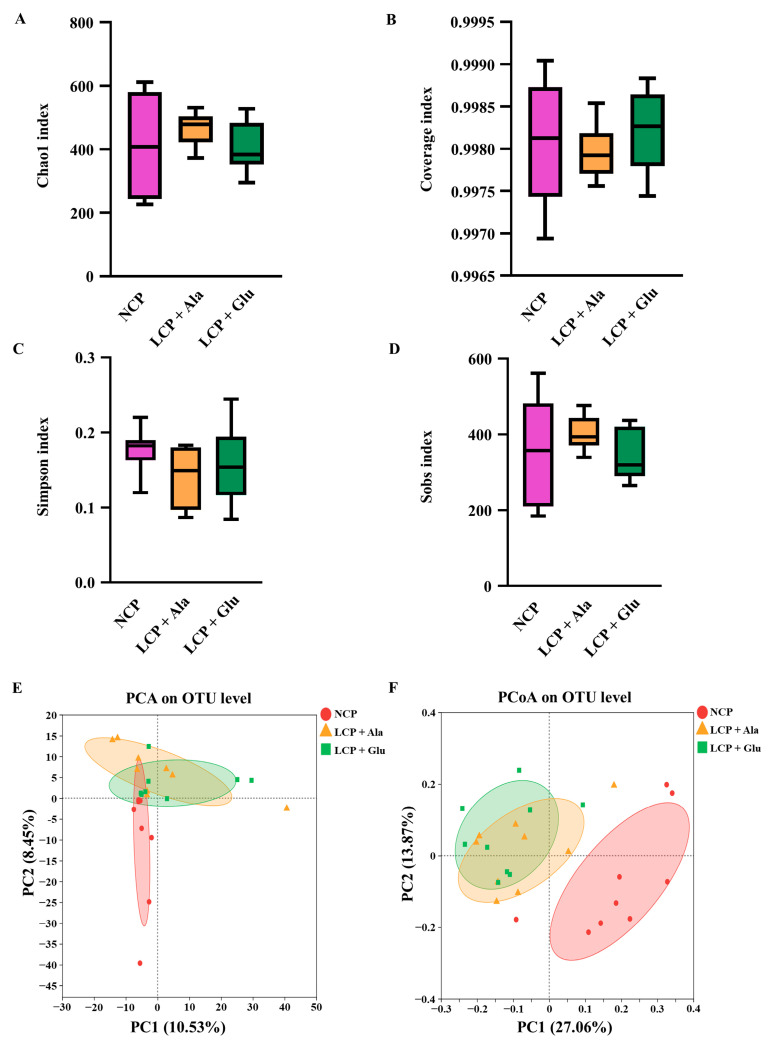
Species composition analysis from Illumina Miseq sequencing. Comparison of α-diversity indices (**A**–**D**) in different dietary groups. PCA (**E**) and PCoA (**F**) results based on OUT levels. NCP = normal crude protein diet; LCP + Ala = low-crude-protein diet supplemented with alanine; LCP + Glu = low-crude-protein diet supplemented with 2.07% glutamate.

**Figure 5 nutrients-17-01465-f005:**
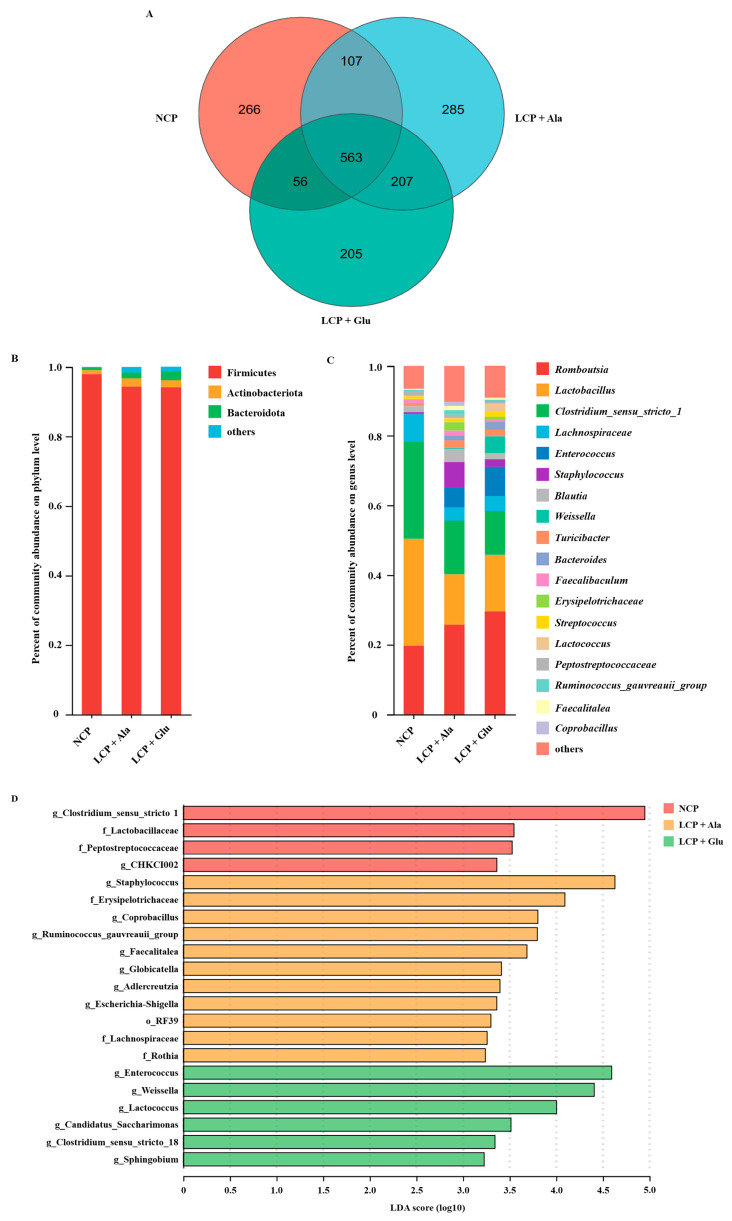
Species diversity analysis and marker species of microbial community in the colon. (**A**) Venn diagram showing unique and shared OTUs. Following a community bar plot analysis, the charts depict the relative abundance of colonic microbiota at phylum (**B**) and genus levels (**C**). (**D**) Histograms of colon microbiota composition in rats of NCP, LCP + Ala, and LCP + Glu group using linear discriminant analysis (LDA) with effect size (LEfSe). NCP = normal crude protein diet; LCP + Ala = low-crude-protein diet supplemented with alanine; LCP + Glu = low-crude-protein diet supplemented with 2.07% glutamate.

**Figure 6 nutrients-17-01465-f006:**
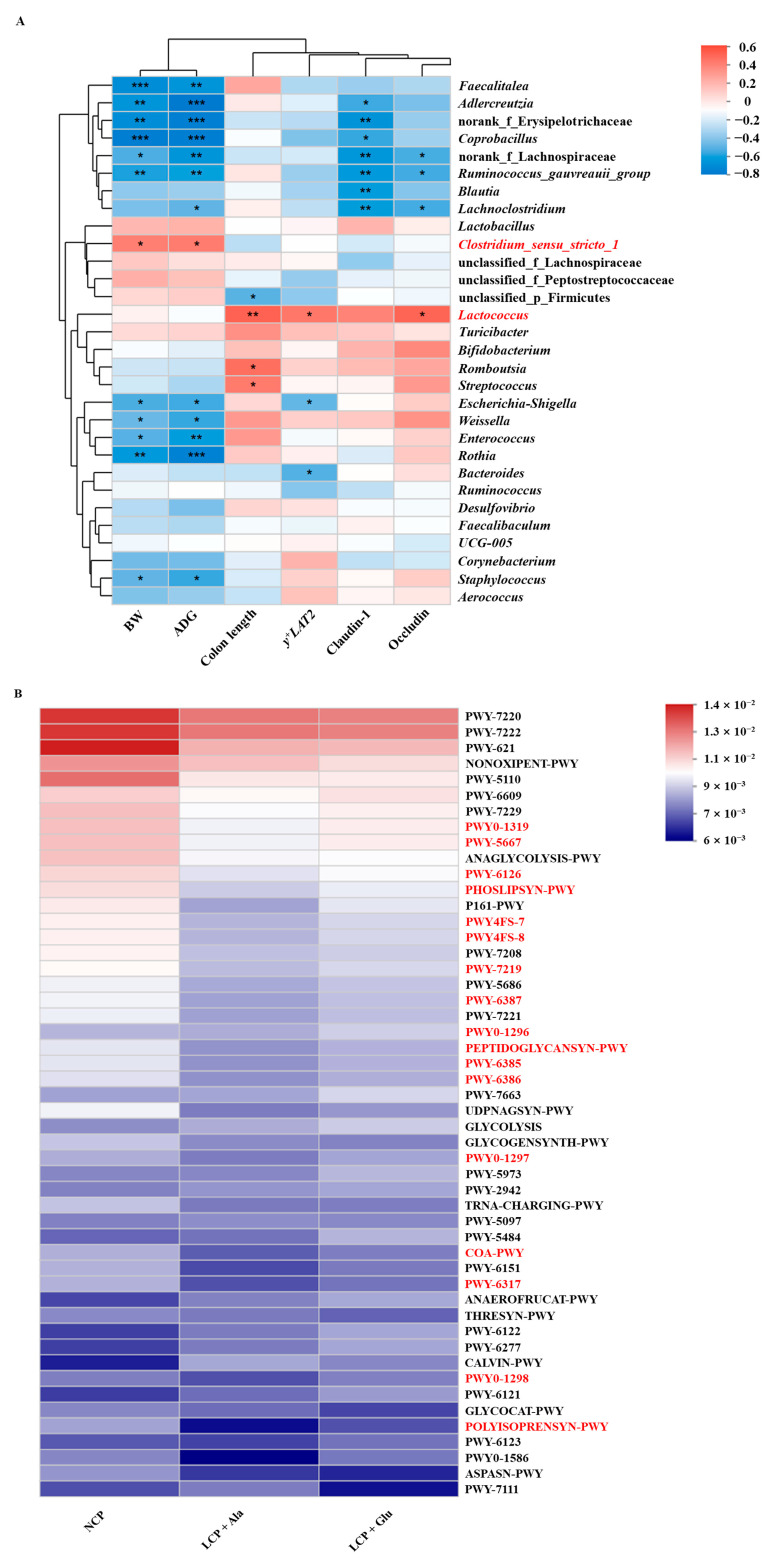
Functional prediction analysis of colon. (**A**) The Spearman correlation analysis of gut microbial composition at genus level with growth performance and intestinal barrier functions of weaned rats. Spearman correlation coefficients of body weight, average daily gain, colon length, and the expression of *y^+^LAT2*, Claudin-1, and Occludin with colonic microbiota are represented by color ranging from red (positive correlation) to blue (negative correlation). *, **, and *** indicate statistically significant differences (*p* < 0.05), (*p* < 0.01), and (*p* < 0.001), respectively. The red fonts indicate bacteria that are significantly and positively correlated with growth performance and intestinal barrier functions. (**B**) The prediction of the functional composition of the gut microbiota between the NCP group, LCP + Ala group, and LCP + Glu group was performed using the phylogenetic investigation of communities by reconstructing the unobserved states (PICRUSt) with bioinformatics software. The red fonts indicate significantly upregulated biosynthesis and utilization pathways. NCP = normal crude protein diet; LCP + Ala = low-crude-protein diet supplemented with alanine; LCP + Glu = low-crude-protein diet supplemented with 2.07% glutamate.

**Figure 7 nutrients-17-01465-f007:**
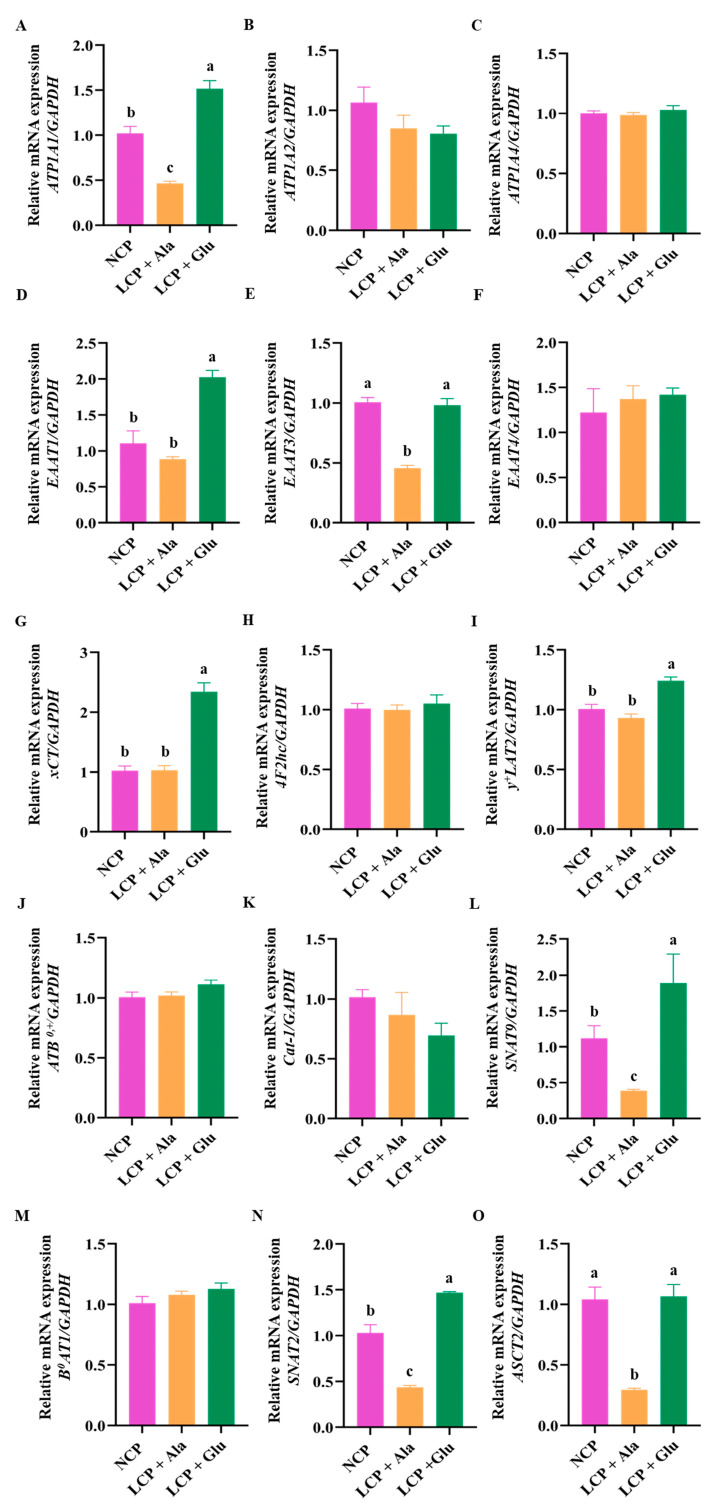
Real-time PCR analysis of Na/K-ATPase (**A**–**C**) and amino acid transporter (**D**–**O**) expression in the liver of 46-day-old weaned rats in the normal protein diet group, the low-protein diet group, and the low-protein diet supplemented with 2.07% glutamate group from days 28 through 46. NCP = normal crude protein diet; LCP + Ala = low-crude-protein diet supplemented with alanine; LCP + Glu = low-crude-protein diet supplemented with 2.07% glutamate. ^a–c^ Different superscripted letters in the same column indicate significant differences (*p* < 0.05).

**Figure 8 nutrients-17-01465-f008:**
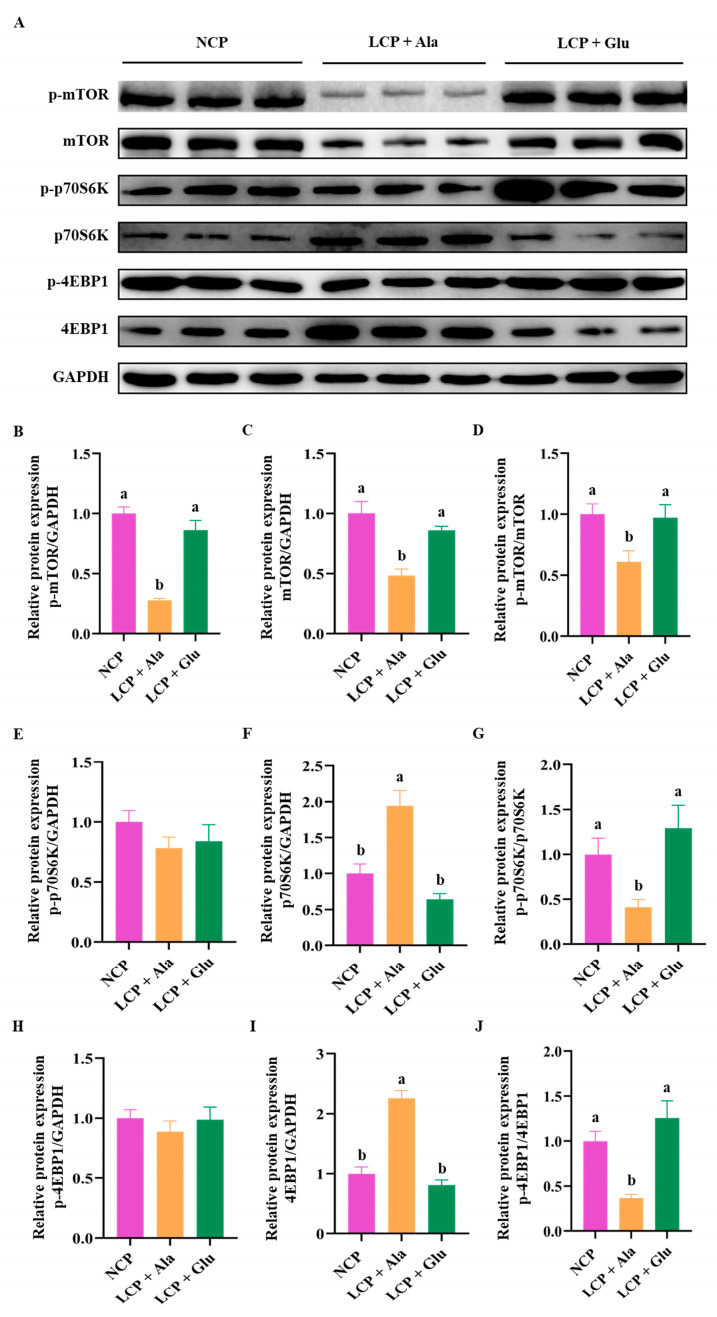
Western blot bands (**A**) and abundances analysis (**B**–**J**) of p-mTOR, mTOR, p-p70S6K, p70S6K, p-4EBP1, and 4EBP1 proteins in the liver of 46-day-old weaned rats in the normal protein diet group, the low-protein diet group, and the low-protein diet supplemented with 2.07% glutamate group from days 28 through 46. NCP = normal crude protein diet; LCP + Ala = low-crude-protein diet supplemented with alanine; LCP + Glu = low-crude-protein diet supplemented with 2.07% glutamate. Representative blots from n = 9 biological replicates are shown. ^a,b^ Different superscripted letters in the same column indicate significant differences (*p* < 0.05).

**Table 1 nutrients-17-01465-t001:** Dietary ingredients and nutrient composition (dry matter basis, %).

Items	NCP	LCP + Ala	LCP + Glu
Ingredients (%)			
Corn starch	39.75	42.64	41.82
Casein	20.00	10.00	10.00
Sucrose	10.00	10.00	10.00
Maltodextrin	13.20	13.20	13.20
Soybean oil	7.00	8.25	8.25
Cellulose	5.00	5.00	5.00
t-Butylhydroquinone	0.0014	0.0014	0.0014
AIN-93G mineral mix ^1^	3.50	3.50	3.50
AIN-93G vitamin mix ^2^	1.00	1.00	1.00
Choline	0.25	0.25	0.25
Lysine	0.00	0.94	0.94
L-Arginine hydrochloride	0.00	0.41	0.41
L-Threonine	0.00	0.41	0.41
L-Tryptophan	0.00	0.12	0.12
DL-Methionine	0.00	0.27	0.27
L-Leucine	0.00	0.89	0.89
L-Isoleucine	0.00	0.49	0.49
L-Valine	0.00	0.61	0.61
L-Histidine	0.00	0.28	0.28
Cystine	0.30	0.00	0.00
Phenylalanine	0.00	0.49	0.49
L-Alanine	0.00	1.25	0.00
L-Glutamate	0.00	0.00	2.07
Calculated nutrient level (%)			
Crude fat	6.98	8.19	8.19
Carbohydrate	61.69	64.52	63.72
Crude protein	18.14	14.52	14.51
Gross energy, kcal/g	4.35	4.35	4.34
Analyzed nutrient level (%)			
Crude fat	7.11	9.61	9.21
Crude protein	18.22	15.41	14.83
Crude ash	2.79	2.67	2.73
Dry matter	86.91	86.95	86.80

NCP = normal crude protein diet; LCP + Ala = low-crude-protein diet supplemented with alanine; LCP + Glu = low-crude-protein diet supplemented with 2.07% glutamate. ^1^ Provided per kilograms of diet: calcium, 5.1 mg; phosphorus, 3.2 mg; potassium, 3.6 mg; magnesium, 0.5 mg; sodium, 1.3 mg; chloride, 2.2 mg; fluorine, 1.0 ppm; iron, 40 ppm; zinc, 35 ppm; manganese, 11 ppm; copper, 6.0 ppm; iodine, 0.21 ppm; chromium, 1.0 ppm; molybdenum, 0.14 ppm; selenium, 0.24 ppm. ^2^ Provided per kg of diet: Vitamin A, 4.0 IU/g; Vitamin D-3 (added), 1.0 IU/g; Vitamin E, 81.6 IU/kg; Vitamin K, 0.75 ppm; Thiamin hydrochloride, 6.1 ppm; Riboflavin, 6.7 ppm; Niacin, 30 ppm; Pantothenic acid, 16 ppm; Folic acid, 2.1 ppm; Pyridoxine, 5.8 ppm; Biotin, 0.2 ppm; Vitamin B-12, 29 mcg/kg; Choline chloride, 1250 ppm.

**Table 2 nutrients-17-01465-t002:** Fecal nitrogen metabolism of 46-day-old weaned rats in the normal protein diet group, the low-protein diet group, and the low-protein diet supplemented with 2.07% glutamate group from days 28 through 46.

Items	1: NCP	2: LCP + Ala	3: LCP + Glu	SEM	*p*-Value		
					1 vs. 2	2 vs. 3	1 vs. 3
Feed intake, g	64.689 ^a1^	52.867 ^b^	53.956 ^b^	3.591	0.071	0.847	0.094
Nitrogen intake, g	1.886 ^a^	1.303 ^b^	1.280 ^b^	0.091	0.005	0.868	0.004
Fecal nitrogen, g	0.232 ^a^	0.164 ^b^	0.110 ^c^	0.006	<0.001	0.001	<0.001
Fecal microbial nitrogen, g	0.035 ^ab^	0.024 ^b^	0.048 ^a^	0.003	0.070	0.004	0.054
Fecal nitrogen/nitrogen intake, %	12.306 ^a^	12.629 ^a^	8.748 ^b^	0.381	0.634	0.001	0.002
Fecal microbial nitrogen/fecal nitrogen, %	15.296 ^b^	14.710 ^b^	44.079 ^a^	2.862	0.918	0.002	0.002
Digestible nitrogen, g	1.654 ^a^	1.139 ^b^	1.170 ^b^	0.086	0.007	0.817	0.009
Apparent nitrogen digestibility, %	87.7 ^b^	87.4 ^b^	91.3 ^a^	0.004	0.598	0.001	0.002

NCP = normal crude protein diet; LCP + Ala = low-crude-protein diet supplemented with alanine; LCP + Glu = low-crude-protein diet supplemented with 2.07% glutamate; SEM = standard error of the mean. ^1 a–c^: Values with different superscripts within a row means they are statistically significant (*p* < 0.05).

## Data Availability

The original contributions presented in the study are included in the article, further inquiries can be directed to the corresponding author.

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
