# Peer review of "Glutamate Supplementation Regulates Nitrogen Metabolism in the Colon and Liver of Weaned Rats Fed a Low-Protein Diet"

_nutrients, 2025, doi:10.3390/nu17091465_

Round 1

Reviewer 1 Report

Comments and Suggestions for Authors

This is a very interesting, well-written paper. There are some issues to be adressed:

  1. How was the amount of the addedd glutamate in the diet chosen? Were there other amounts tried?
  2. In which form glutamate and alanine were added in the diet? Why alanine was chosen and not another amino acid?
  3. The exact nutritional and energy content of the diets is not presented. The exact amounts of individual amino acids provided by the diets are not presented.
  4. The possible adverse effects of chronic glutamate administration should be discussed. 
  5. How can these results have beneficial effects for humans? Which amount of glutamate should be consumed?
  6. Are there other amino acids (i.e. aspartate) that could possibly also have beneficial effects and should be discussed in the paper?

Author Response

Comment 1: How was the amount of the added glutamate in the diet chosen? Were there other amounts tried?

Response 1:

Thank you for your valuable comments. We added 2.07% glutamate to the LCP + Glu group with the aim of aligning its dietary glutamate nutritional level with that of the control group (4.12%). To facilitate a clearer understanding of this adjustment for you and other readers, we have included Table 1 and Table S1 in the manuscript. These tables provide detailed information on the glutamate nutrient levels in the diets of all treatment groups. As shown in the data, the diet of NCP group contained 4.12% glutamate, whereas the of LCP + Ala group diet contained only 2.06%. By supplementing the diet of LCP + Glu group with 99.5%-pure glutamate (2.07%), we successfully raised its glutamate nutritional level to 4.12%, achieving numerical parity with the control group.

Thanks again for your advice. We have added specific details regarding the amount of glutamate supplemented in the LCP + Glu group in lines 130-132, 145148, and 154156 of the manuscript to further emphasize this key point. The details are as follows: “a normal crude protein diet (NCP) group, a low crude protein diet supplemented with alanine (LCP + Ala), or a low crude protein diet supplemented with 2.07% glutamate (LCP + Glu).

Additionally, the glutamate content was 4.12% in the NCP group and 2.06% in the LCP + Ala group. After adding 2.07% glutamate (purity 99.5%), the glutamate content in the LCP + Glu group diet increased to 4.12%, aligning with the dietary glutamate content of NCP group.

The feed was provided by Xiao Shu You Tai (Beijing) Biotechnology Co., Ltd., and de-tailed information on ingredients and nutrients is shown in Table 1 and Table S1.”

In addition, we have not yet experimented with other doses of glutamate supplementation. We appreciate your valuable suggestions and will consider investigating the effects of glutamate supplementation at varying doses in future studies.

Table 1. Dietary ingredients and nutrient composition (Dry matter basis, %).

Items

NCP

LCP + Ala

LCP + Glu

Ingredients (%)

Corn starch

39.75

42.64

41.82

Casein

20.00

10.00

10.00

Sucrose

10.00

10.00

10.00

Maltodextrin

13.20

13.20

13.20

Soybean oil

7.00

8.25

8.25

Cellulose

5.00

5.00

5.00

t-Butylhydroquinone

0.0014

0.0014

0.0014

AIN-93G mineral mix 1

3.50

3.50

3.50

AIN-93G vitamin mix 2

1.00

1.00

1.00

Choline

0.25

0.25

0.25

Lysine

0.00

0.94

0.94

L-Arginine hydrochloride

0.00

0.41

0.41

L-Threonine

0.00

0.41

0.41

L-Tryptophan

0.00

0.12

0.12

DL-Methionine

0.00

0.27

0.27

L-Leucine

0.00

0.89

0.89

L-Isoleucine

0.00

0.49

0.49

L-Valine

0.00

0.61

0.61

L-Histidine

0.00

0.28

0.28

Cystine

0.30

0.00

0.00

Phenylalanine

0.00

0.49

0.49

L-Alanine

0.00

1.25

0.00

L-Glutamate

0.00

0.00

2.07

Calculated nutrient level (%)

Crude fat

6.98

8.19

8.19

Carbohydrate

61.69

64.52

63.72

Crude protein

18.14

14.52

14.51

Gross energy, kcal/g

4.35

4.35

4.34

Analyzed nutrient level (%)

Crude fat

7.11

9.61

9.21

Crude protein

18.22

15.41

14.83

Crude ash

2.79

2.67

2.73

Dry matter

86.91

86.95

86.80

NCP = normal crude protein diet, LCP + Ala = low crude protein diet supplemented with alanine, LCP + Glu = low crude protein diet supplemented with 2.07% glutamate.

1 Provided per kilograms of diet: Calcium, 5.1 mg; Phosphorus, 3.2 mg; Potassium, 3.6 mg; Magnesium, 0.5 mg; Sodium, 1.3 mg; Chloride, 2.2 mg; Fluorine, 1.0 ppm; Iron, 40 ppm; Zinc, 35 ppm; Manganese, 11 ppm; Copper, 6.0 ppm; Iodine, 0.21 ppm; Chromium, 1.0 ppm; Molybdenum, 0.14 ppm; Selenium, 0.24 ppm.

2 Provided per kg of diet: Vitamin A, 4.0 IU/g; Vitamin D-3 (added), 1.0 IU/g; Vitamin E, 81.6 IU/kg; Vitamin K, 0.75 ppm; Thiamin hydrochloride, 6.1 ppm; Riboflavin, 6.7 ppm; Niacin, 30 ppm; Pantothenic acid, 16 ppm; Folic acid, 2.1 ppm; Pyridoxine, 5.8 ppm; Biotin, 0.2 ppm; Vitamin B-12, 29 mcg/kg; Choline chloride, 1250 ppm.

Table S1. Dietary amino acid composition (Dry matter basis, %).

Items

Calculated nutrient level (%)

Analyzed nutrient level (%)

NCP

LCP + Ala

LCP + Glu

NCP

LCP + Ala

LCP + Glu

Essential amino acids

Arginine

0.68

0.68

0.68

0.69

0.69

0.68

Histidine

0.56

0.56

0.56

0.55

0.55

0.56

Isoleucine

0.98

0.98

0.98

1.03

1.04

0.99

Leucine

1.76

1.76

1.76

1.74

1.76

1.77

Lysine

1.50

1.50

1.50

1.46

1.45

1.46

Methionine

0.53

0.53

0.53

0.54

0.55

0.55

Phenylalanine

0.97

0.97

0.97

0.93

0.94

0.90

Threonine

0.82

0.82

0.82

0.77

0.81

0.80

Tryptophan

0.24

0.24

0.24

0.27

0.28

0.27

Valine

1.21

1.21

1.21

1.19

1.19

1.19

Nonessential amino acids

Alanine

0.55

1.52

0.28

0.53

1.55

0.29

Aspartic acid

1.29

0.64

0.64

1.31

0.67

0.68

Cystine

0.38

0.04

0.04

0.36

0.03

0.03

Glutamate

4.12

2.06

4.12

4.14

2.08

4.12

Glycine

0.37

0.19

0.19

0.39

0.16

0.17

Proline

2.16

1.08

1.08

2.20

1.11

1.02

Serine

1.02

0.51

0.51

1.04

0.51

0.53

Tyrosine

1.01

0.51

0.51

1.05

0.54

0.53

EAA + NEAA

20.15

15.80

16.62

20.18

15.91

16.54

Comment 2: In which form glutamate and alanine were added in the diet? Why alanine was chosen and not another amino acid?

Response 2:

Thank you for raising this important question. In our experimental design, we used L-glutamate crystals and L-alanine crystals with a purity of 99.5% as dietary raw materials to prepare the diets. The primary reason for selecting alanine is that, as a neutral amino acid, it effectively regulates the iso-nitrogen content in the diet, thereby ensuring nitrogen-level consistency between the LCP + Ala group and the LCP + Glu group. This approach was intended to eliminate any potential interference from nitrogen differences in the experimental results, allowing us to focus specifically on the functional characteristics of glutamate (rather than its nitrogen contribution).

Thank you again for your valuable comments. We have filled in the relevant details in lines 148154 of the manuscript, as follows: “Meanwhile, alanine was selected as it is a neutral amino acid suitable for regulating iso-nitrogen levels in the diet. This ensured the iso-nitrogen characteristics of both the LCP + Ala and LCP + Glu groups, thereby excluding any potential interference from nitrogen content differences and allowing us to focus on the functional specificity of glutamate (rather than its nitrogen contribution).”

Comment 3: The exact nutritional and energy content of the diets is not presented. The exact amounts of individual amino acids provided by the diets are not presented.

Response 3:

We fully concur with your suggestion. Consequently, Table 1 has been incorporated into the manuscript to provide a detailed overview of the precise nutrient composition, energy levels, and individual amino acid concentrations in the diet.

Thanks to your valuable advice, we have provided a detailed description of the specific contents of Table 1 in lines 154156 of the manuscript. The details are presented as follows: “The feed was provided by Xiao Shu You Tai (Beijing) Biotechnology Co., Ltd., and de-tailed information on ingredients and nutrients is shown in Table 1 and Table S1.”

Table 1. Dietary ingredients and nutrient composition (Dry matter basis, %).

Items

NCP

LCP + Ala

LCP + Glu

Ingredients (%)

Corn starch

39.75

42.64

41.82

Casein

20.00

10.00

10.00

Sucrose

10.00

10.00

10.00

Maltodextrin

13.20

13.20

13.20

Soybean oil

7.00

8.25

8.25

Cellulose

5.00

5.00

5.00

t-Butylhydroquinone

0.0014

0.0014

0.0014

AIN-93G mineral mix 1

3.50

3.50

3.50

AIN-93G vitamin mix 2

1.00

1.00

1.00

Choline

0.25

0.25

0.25

Lysine

0.00

0.94

0.94

L-Arginine hydrochloride

0.00

0.41

0.41

L-Threonine

0.00

0.41

0.41

L-Tryptophan

0.00

0.12

0.12

DL-Methionine

0.00

0.27

0.27

L-Leucine

0.00

0.89

0.89

L-Isoleucine

0.00

0.49

0.49

L-Valine

0.00

0.61

0.61

L-Histidine

0.00

0.28

0.28

Cystine

0.30

0.00

0.00

Phenylalanine

0.00

0.49

0.49

L-Alanine

0.00

1.25

0.00

L-Glutamate

0.00

0.00

2.07

Calculated nutrient level (%)

Crude fat

6.98

8.19

8.19

Carbohydrate

61.69

64.52

63.72

Crude protein

18.14

14.52

14.51

Gross energy, kcal/g

4.35

4.35

4.34

Analyzed nutrient level (%)

Crude fat

7.11

9.61

9.21

Crude protein

18.22

15.41

14.83

Crude ash

2.79

2.67

2.73

Dry matter

86.91

86.95

86.80

NCP = normal crude protein diet, LCP + Ala = low crude protein diet supplemented with alanine, LCP + Glu = low crude protein diet supplemented with 2.07% glutamate, SEM = standard error of the mean.

1 Provided per kilograms of diet: Calcium, 5.1 mg; Phosphorus, 3.2 mg; Potassium, 3.6 mg; Magnesium, 0.5 mg; Sodium, 1.3 mg; Chloride, 2.2 mg; Fluorine, 1.0 ppm; Iron, 40 ppm; Zinc, 35 ppm; Manganese, 11 ppm; Copper, 6.0 ppm; Iodine, 0.21 ppm; Chromium, 1.0 ppm; Molybdenum, 0.14 ppm; Selenium, 0.24 ppm.

2 Provided per kg of diet: Vitamin A, 4.0 IU/g; Vitamin D-3 (added), 1.0 IU/g; Vitamin E, 81.6 IU/kg; Vitamin K, 0.75 ppm; Thiamin hydrochloride, 6.1 ppm; Riboflavin, 6.7 ppm; Niacin, 30 ppm; Pantothenic acid, 16 ppm; Folic acid, 2.1 ppm; Pyridoxine, 5.8 ppm; Biotin, 0.2 ppm; Vitamin B-12, 29 mcg/kg; Choline chloride, 1250 ppm.

Comment 4: The possible adverse effects of chronic glutamate administration should be discussed.

Response 4: We sincerely appreciate your valuable suggestions. First, since glutamate shares the same transporter system with other amino acids, long-term ingestion of glutamate may increase its availability in the body, thereby affecting the efficiency of transport of other amino acids across cell membranes and their distribution in tissues through competitive inhibition mechanisms. Second, long-term supplementation of exogenous glutamate might exert a negative feedback effect on the endogenous glutamate synthesis pathway, potentially weakening the capacity for endogenous glutamate production. Third, given the significant immunomodulatory properties of glutamate, it is essential to evaluate the impact of long-term glutamate consumption on immune system homeostasis. Fourth, considering the regulatory role of glutamate in the nervous system, further investigation is required to determine whether long-term glutamate supplementation could induce potential adverse effects on nervous system development or function. Fifth, due to the organism's adaptive response to glutamate supplementation, abrupt cessation of glutamate intake might lead to a sharp decline in glutamate levels within the body, thereby increasing the risk of health issues associated with glutamate deficiency. In summary, based on the potential changes in amino acid transport, glutamate metabolic disorders, immune system imbalance, abnormal nervous system development, and negative effects following the cessation of supplementation, we recommend adopting short-term, intermittent glutamate supplementation strategies to effectively mitigate the risks associated with long-term glutamate consumption. Additionally, we highly value your advice and plan to conduct subsequent studies to comprehensively explore the effectiveness, potential toxicity, and physiological consequences of long-term glutamate supplementation.

Once again, thank you for highlighting this critical issue. The detailed discussion mentioned above has been incorporated into lines 608615 of the manuscript. “In addition, given the potential changes in amino acid transport, glutamate metabolism, immune system function, nervous system development, as well as the adverse effects associated with discontinuing glutamate supplementation [105,106], we recommend short-term and intermittent multiple administrations of glutamate supplementation to minimize the risks associated with long-term consumption. Furthermore, additional trials are warranted to evaluate the efficacy and safety of long-term glutamate supplementation, including its potential toxicity and the physiological impacts following cessation after prolonged use.”

Comment 5: How can these results have beneficial effects for humans? Which amount of glutamate should be consumed?

Response 5:

Thank you for your valuable advice. According to the experimental results in rats, incorporating glutamate (2.07%) into a low-protein diet significantly promotes the growth and development of post-weaning rats. Based on this finding, we hypothesize that for human children, particularly those in developing countries with insufficient protein intake, daily supplementation of approximately 2-4 grams of glutamate may potentially help alleviate malnutrition. However, further research is necessary to validate the safety and practical application effects of such supplementation.

We consider your suggestions to be highly constructive. As a result, we have incorporated the relevant content into the manuscript, specifically from lines 621628. The specific content is as follows: “Specifically, according to the experimental results in rats, incorporating 2.07% gluta-mate into a low-protein diet significantly enhances the growth and development of post-weaning rats. If this insight is extended to human children, particularly those in developing countries with insufficient protein intake, it is hypothesized that daily supplementation of approximately 2-4 grams of glutamate may offer potential benefits in addressing malnutrition. Nevertheless, the safety and practical efficacy of such supplementation must be rigorously evaluated through additional human trials.”

Comment 6: Are there other amino acids (i.e. aspartate) that could possibly also have beneficial effects and should be discussed in the paper?

Response 6:

Agreed. We sincerely appreciate your valuable advice. At the same time, in addition to glutamate, amino acids such as aspartate, arginine, and glutamine may also enhance growth performance under low-protein diet conditions by regulating nitrogen metabolism, promoting protein synthesis, and modulating immune function.

Specifically, aspartate, as an acidic amino acid like glutamate, is involved in the urea cycle and the tricarboxylic acid cycle. It indirectly improves growth performance by regulating energy metabolism, lipid metabolism, and intestinal microbiota structure. However, since aspartate and glutamate share the same transporter system, metabolic competition between them may occur, potentially affecting their complementary effects.

Arginine is a conditionally essential amino acid with nutritional significance for piglets, particularly when endogenous synthesis is insufficient. Studies have demonstrated that dietary supplementation with arginine can significantly enhance the immune response of weaned piglets and exert immunomodulatory effects by promoting nitric oxide (NO) synthesis. Nevertheless, high doses of arginine may lead to excessive NO production, thereby increasing the risk of intestinal inflammation. Therefore, it is crucial to appropriately balance its supplemental dose.

Glutamine, as a conditionally essential amino acid, serves not only as the precursor of glutamate but also significantly improves the growth performance, serum biochemical parameters, and antioxidant capacity of weaned piglets under low-protein diet conditions. However, given that glutamine is readily catabolized in vivo, the effectiveness and stability of exogenous supplementation require further evaluation.

In summary, although this study revealed that glutamate significantly enhances the growth of rats on a low-protein diet, amino acids such as aspartate and arginine may also play synergistic roles by regulating nitrogen metabolism, promoting protein synthesis, and supporting immune function. Future studies should systematically compare the intervention effects of different amino acids and explore cost-effective, safe, and efficient combination strategies to optimize nutritional intervention programs for malnourished individuals.

Thank you again for your insightful suggestions, which have been incorporated into lines 576600 of the manuscript. “Meanwhile, in addition to glutamate, amino acids such as aspartate, arginine, and glutamine may contribute to improved growth performance through the regulation of nitrogen metabolism and enhancement of immune function. Amino acids such as aspartate, arginine, and glutamine in low-protein diets may contribute to improved growth performance through the regulation of nitrogen metabolism and enhancement of immune function. Aspartate and glutamate are both acidic amino acids that participate in the urea cycle and the tricarboxylic acid (TCA) cycle, playing crucial roles in energy metabolism. Furthermore, aspartate can modulate energy and lipid metabolism by regulating immune function and optimizing the composition of gut microbiota, there-by potentially enhancing growth performance [100,101]. However, aspartate and glutamate exhibit metabolic competition, such as sharing transporters, which could potentially influence their supplementation efficacy [44]. Due to insufficient endogenous synthesis, arginine is considered an essential amino acid for piglets. Dietary supplementation with arginine has been reported to strengthen the immune response of weaned piglets [102,103]. Nevertheless, high doses of argi-nine may exacerbate intestinal inflammation by increasing nitric oxide production, necessitating careful dose optimization. Glutamine, classified as a conditionally essential amino acid, serves as a precursor to glutamate. It can improve the growth performance of weaned piglets on low-protein diets while enhancing serum physicochemical parameters and antioxidant capacity [104]. Despite its benefits, glutamine is prone to rapid degradation in vivo, raising questions about the effectiveness of exogenous supplementation. In summary, although this study demonstrated that glutamate significantly enhances the growth of rats on low-protein diets, amino acids like aspartate and arginine may also contribute synergistically by regulating nitrogen metabolism and promoting protein synthesis.

Future studies should systematically assess the effects of various amino acid interventions, such as incorporating low-protein diets supplemented with aspartate, arginine, or glutamine. Additionally, introducing a glutamate deprivation model (e.g., using glutamate transporter inhibitors) could help observe whether growth inhibition and nitrogen metabolism disturbances can be reversed by exogenous glutamate supplementation. Simultaneously, it is essential to explore cost-effective and safe combi-nation strategies to further enhance nutritional intervention programs for undernourished populations.”

Reviewer 2 Report

Comments and Suggestions for Authors

Thank you for the opportunity to review the paper submitted by Jiang et al to Nutrients. Please see below my comments on the above manuscript:

-No control is used to specifically determine whether the outcome of this study is attributed to glutamate. In other words, how can we make sure not similar responses can be obtained by supplementing other amino acids and the results seen here are unique for that amino acid? To be able to answer to that question, another control group should have been included in the experimental design with either supplementation of other amino acids (Ala used in LCP diet cannot help as that’s a neutral amino acid used for adjusting N content) or depleting glutamic acid.

-Experimental details are missing. The experimental diets ingredients and composition are not given. It’s not clear why 2.07% glutamate is selected to be supplemented to low crude protein diet. Further, not clear whether all three diets were isocaloric or not. What was the justification for using Ala for adjusting N and that adding Ala has no negative effects on nitrogen metabolism in LCP diet. What made the authors to decide on 18 days of experimental period based on changes in body weight? (lines 101-103)

-It’s not clear why authors are focused on colon in this study. From the background information given in the introduction (line 43-53), and the objective of the study (lines 81-83), it seems that the colon is hypothesized to be a secondary site for N metabolism following glutamic acid supplementation. However, when it comes to analysis, none of uric acid cycle genes that are critical for nitrogen metabolism have been analyzed in colon and the analysis are limited to morphology, amino acid transports, and tight junction markers. It is confusing why colon is used and not other segments are analyzed for the above parameters. For gut microbiota analysis, it would have been more informative if cecal contents were used as rats are cecal fermenters not colon fermenters.

-For immunoblot analysis, it seems that only n of 3 is used (see Fig. 3 and Fig. 8) while it’s previously known that there is a large variation in protein expression of target molecules in vivo and at least n of 6 is required for a valid outcome

Author Response

Comment 1: No control is used to specifically determine whether the outcome of this study is attributed to glutamate. In other words, how can we make sure not similar responses can be obtained by supplementing other amino acids and the results seen here are unique for that amino acid? To be able to answer to that question, another control group should have been included in the experimental design with either supplementation of other amino acids (Ala used in LCP diet cannot help as that’s a neutral amino acid used for adjusting N content) or depleting glutamic acid.

Response 1:

Thank you for this important question. We fully agree that the effects of glutamate supplementation in a low-protein model may interact with other amino acids, necessitating validation of specificity through a well-designed control. Below is our explanation and additional analysis:

First, we selected alanine as an iso-nitrogen control to focus on the functional specificity of glutamate rather than its nitrogen contribution, by eliminating interference from pure "nitrogen supplementation." The impact of alanine on mammalian growth performance and nitrogen metabolism is minimal, as confirmed by multiple studies. For instance, some studies have demonstrated that alanine supplementation in low-protein models did not significantly enhance growth or improve nitrogen metabolic function, further supporting the rationale for using alanine as a control.

Secondly, experimental evidence indicates that the LCP + Ala group failed to activate the mTOR pathway, suggesting that glutamate exerts specific effects on nitrogen metabolism in the colon and liver via distinct signaling pathways, rather than relying solely on nitrogen supplementation.

Additionally, this study did not include supplementation groups for other amino acids (e.g., aspartate, arginine, and glutamine), making it impossible to entirely rule out similar effects from other acidic or metabolically active amino acids. This limitation arises from the experimental scale and core objective of focusing on glutamate. However, literature review reveals that, besides glutamate, amino acids such as aspartate, arginine, and glutamine may also improve growth performance under low-protein conditions by regulating nitrogen metabolism and enhancing immune function. Nevertheless, it remains to be further investigated whether supplementation with aspartate, arginine, or glutamine in low-protein diets fortified with 10 essential amino acids can elicit comparable effects to glutamate administration, particularly regarding three critical aspects: modulation of colonic mechanical and mucosal barrier functions, regulation of gut microbiota to mitigate nitrogen emissions, and activation of hepatic mTOR signaling pathways coupled with enhanced amino acid metabolism.

In response to your valuable suggestion, we plan to include low-protein + aspartate/arginine/glutamine groups in future studies to directly compare the effects of various amino acid interventions. Furthermore, we will introduce a glutamate deprivation model (e.g., using glutamate transporter inhibitors) to examine whether growth inhibition and nitrogen metabolism suppression can be reversed by exogenous glutamate addition. Simultaneously, we aim to explore cost-effective and safe combination strategies to optimize nutritional programs for undernourished populations.

Once again, thank you for your suggestion. The relevant content has been added to lines 576600. “Meanwhile, in addition to glutamate, amino acids such as aspartate, arginine, and glutamine may contribute to improved growth performance through the regulation of nitrogen metabolism and enhancement of immune function. Amino acids such as aspartate, arginine, and glutamine in low-protein diets may contribute to improved growth performance through the regulation of nitrogen metabolism and enhancement of immune function. Aspartate and glutamate are both acidic amino acids that participate in the urea cycle and the tricarboxylic acid cycle, playing crucial roles in energy metabolism. Furthermore, aspartate can modulate energy and lipid metabolism by regulating immune function and optimizing the composition of gut microbiota, thereby potentially enhancing growth performance [100,101]. However, Aspartate and glutamate exhibit metabolic competition, such as sharing transporters, which could potentially influence their supplementation efficacy [44]. Due to insufficient endogenous synthesis, arginine is considered an essential amino acid for piglets. Dietary supplementation with arginine has been reported to strengthen the immune response of weaned piglets [102,103]. Nevertheless, high doses of arginine may exacerbate intestinal inflammation by increasing nitric oxide production, necessitating careful dose optimization. Glutamine, classified as a conditionally essential amino acid, serves as a precursor to glutamate. It can improve the growth performance of weaned piglets on low-protein diets while enhancing serum physicochemical parameters and antioxidant capacity [104]. Despite its benefits, glutamine is prone to rapid degradation in vivo, raising questions about the effectiveness of exogenous supplementation. In summary, although this study demonstrated that glutamate significantly enhances the growth of rats on low-protein diets, amino acids like aspartate and arginine may also contribute synergistically by regulating nitrogen metabolism and promoting protein synthesis.

Future studies should systematically assess the effects of various amino acid interventions, such as incorporating low-protein diets supplemented with aspartate, arginine, or glutamine. Additionally, introducing a glutamate deprivation model (e.g., using glutamate transporter inhibitors) could help observe whether growth inhibition and nitrogen metabolism disturbances can be reversed by exogenous glutamate supplementation. Simultaneously, it is essential to explore cost-effective and safe combination strategies to further enhance nutritional intervention programs for undernourished populations”

Comment 2: Experimental details are missing. The experimental diets ingredients and composition are not given. It’s not clear why 2.07% glutamate is selected to be supplemented to low crude protein diet. Further, not clear whether all three diets were isocaloric or not. What was the justification for using Ala for adjusting N and that adding Ala has no negative effects on nitrogen metabolism in LCP diet. What made the authors to decide on 18 days of experimental period based on changes in body weight? (lines 101-103)

Response 2:

Thank you for pointing this out. We have added the complete diet formulation table in the revised draft, and Table 1 and Table S1 detail the raw material composition and nutrient levels (including crude protein, crude fat, carbohydrate, amino acid content, and total energy) of the normal crude protein group (NCP), low crude protein diet supplemented with alanine group (LCP + Ala), and low crude protein diet supplemented with 2.07% glutamate group (LCP + Glu).

The dosage of 2.07% glutamate is based on the principle of equal compensation. Specifically, the glutamate content in the NCP diet was 4.12% (derived from casein), while the glutamate content in the LCP + Ala diet decreased to 2.06% due to reduced protein content. By supplementing 2.07% glutamate (purity 99.5%), the glutamate level in the LCP + Glu group was restored to that of the NCP group.

All diets were designed with an isocaloric structure (total energy of 4.35 kcal/g for all three treatment groups). The LCP + Ala diet and LCP + Glu diet compensated for the calories lost due to reduced protein by increasing the proportion of carbohydrates (cornstarch), ensuring consistent total energy across all groups. This design effectively eliminates potential interference of energy differences on growth performance.

The reasons for selecting alanine as the iso-nitrogen control are as follows: First, from the perspective of nitrogen balance, the supplemental amount of alanine in the LCP + Ala group (2.15%) ensured that its total nitrogen content matched that of the LCP + Glu group (2.07% glutamate nitrogen content = 2.15% alanine nitrogen content). Second, alanine is metabolically inert and does not significantly affect the nitrogen metabolism function of the low-protein diet. Additionally, numerous studies have confirmed that alanine is suitable as an iso-nitrogen control, and in low-protein models, alanine supplementation typically does not exhibit growth-promoting effects or significant nitrogen metabolism regulation. Therefore, we selected alanine to exclude the influence of nitrogen content on the experimental results. 

The experimental period was set to 18 days based on the following considerations: First, the rapid growth phase of weaned rats. Weaned rats aged 28 days are in a stage of rapid weight gain (weight doubling) within 2-3 weeks, which is particularly sensitive to nutritional interventions. Second, consistency with the literature. Our design aligns with similar studies that typically assess the effects of low-protein diets over 14-21 days. Third, balancing statistical power and ethical requirements. In this experiment, the weight difference between the LCP + Ala group and the LCP + Glu group reached a highly significant level at 18 days. To balance statistical power and minimize animal use time, the experimental period was ultimately determined to be 18 days. 

Thank you for your attention to detail. We have supplemented the dietary ingredient list and clarified the scientific basis for glutamate dosage, isocaloric design, alanine control, and experimental period. These designs are supported by the literature to ensure the reliability of the conclusions. The revised version will further clarify these details to enhance the transparency of the paper. The relevant contents have been added to lines 134143 and 153154 of the manuscript, as follows: “The experimental period in this study was set to 18 days. This duration was chosen because rats weaned at 28 days of age enter a rapid growth phase over the subsequent 2-3 weeks (during which their weight doubles), making them particularly sensitive to nutritional interventions [42]. Additionally, previous studies have utilized a time window of 14-21 days to assess the effects of low-protein diets [43-46], and the design of this study aligns with these durations. Furthermore, by day 18 of the experiment, a highly significant difference in body weight was observed between the LCP + Ala group and the LCP + Glu group. Considering both statistical validity and ethical concerns (minimizing animal use time), the experimental period was ultimately determined to be 18 days.

In addition, all diets were designed to be isocaloric, providing a consistent total energy content of 4.35 kcal/g across all three treatment groups.”

Table 1. Dietary ingredients and nutrient composition (Dry matter basis, %).

Items

NCP

LCP + Ala

LCP + Glu

Ingredients (%)

Corn starch

39.75

42.64

41.82

Casein

20.00

10.00

10.00

Sucrose

10.00

10.00

10.00

Maltodextrin

13.20

13.20

13.20

Soybean oil

7.00

8.25

8.25

Cellulose

5.00

5.00

5.00

t-Butylhydroquinone

0.0014

0.0014

0.0014

AIN-93G mineral mix 1

3.50

3.50

3.50

AIN-93G vitamin mix 2

1.00

1.00

1.00

Choline

0.25

0.25

0.25

Lysine

0.00

0.94

0.94

L-Arginine hydrochloride

0.00

0.41

0.41

L-Threonine

0.00

0.41

0.41

L-Tryptophan

0.00

0.12

0.12

DL-Methionine

0.00

0.27

0.27

L-Leucine

0.00

0.89

0.89

L-Isoleucine

0.00

0.49

0.49

L-Valine

0.00

0.61

0.61

L-Histidine

0.00

0.28

0.28

Cystine

0.30

0.00

0.00

Phenylalanine

0.00

0.49

0.49

L-Alanine

0.00

1.25

0.00

L-Glutamate

0.00

0.00

2.07

Calculated nutrient level (%)

Crude fat

6.98

8.19

8.19

Carbohydrate

61.69

64.52

63.72

Crude protein

18.14

14.52

14.51

Gross energy, kcal/g

4.35

4.35

4.34

Analyzed nutrient level (%)

Crude fat

7.11

9.61

9.21

Crude protein

18.22

15.41

14.83

Crude ash

2.79

2.67

2.73

Dry matter

86.91

86.95

86.80

NCP = normal crude protein diet, LCP + Ala = low crude protein diet supplemented with alanine, LCP + Glu = low crude protein diet supplemented with 2.07% glutamate.

1 Provided per kilograms of diet: Calcium, 5.1 mg; Phosphorus, 3.2 mg; Potassium, 3.6 mg; Magnesium, 0.5 mg; Sodium, 1.3 mg; Chloride, 2.2 mg; Fluorine, 1.0 ppm; Iron, 40 ppm; Zinc, 35 ppm; Manganese, 11 ppm; Copper, 6.0 ppm; Iodine, 0.21 ppm; Chromium, 1.0 ppm; Molybdenum, 0.14 ppm; Selenium, 0.24 ppm.

2 Provided per kg of diet: Vitamin A, 4.0 IU/g; Vitamin D-3 (added), 1.0 IU/g; Vitamin E, 81.6 IU/kg; Vitamin K, 0.75 ppm; Thiamin hydrochloride, 6.1 ppm; Riboflavin, 6.7 ppm; Niacin, 30 ppm; Pantothenic acid, 16 ppm; Folic acid, 2.1 ppm; Pyridoxine, 5.8 ppm; Biotin, 0.2 ppm; Vitamin B-12, 29 mcg/kg; Choline chloride, 1250 ppm.

Table S1. Dietary amino acid composition (Dry matter basis, %).

Items

Calculated nutrient level (%)

Analyzed nutrient level (%)

NCP

LCP + Ala

LCP + Glu

NCP

LCP + Ala

LCP + Glu

Essential amino acids

Arginine

0.68

0.68

0.68

0.69

0.69

0.68

Histidine

0.56

0.56

0.56

0.55

0.55

0.56

Isoleucine

0.98

0.98

0.98

1.03

1.04

0.99

Leucine

1.76

1.76

1.76

1.74

1.76

1.77

Lysine

1.50

1.50

1.50

1.46

1.45

1.46

Methionine

0.53

0.53

0.53

0.54

0.55

0.55

Phenylalanine

0.97

0.97

0.97

0.93

0.94

0.90

Threonine

0.82

0.82

0.82

0.77

0.81

0.80

Tryptophan

0.24

0.24

0.24

0.27

0.28

0.27

Valine

1.21

1.21

1.21

1.19

1.19

1.19

Nonessential amino acids

Alanine

0.55

1.52

0.28

0.53

1.55

0.29

Aspartic acid

1.29

0.64

0.64

1.31

0.67

0.68

Cystine

0.38

0.04

0.04

0.36

0.03

0.03

Glutamate

4.12

2.06

4.12

4.14

2.08

4.12

Glycine

0.37

0.19

0.19

0.39

0.16

0.17

Proline

2.16

1.08

1.08

2.20

1.11

1.02

Serine

1.02

0.51

0.51

1.04

0.51

0.53

Tyrosine

1.01

0.51

0.51

1.05

0.54

0.53

EAA + NEAA

20.15

15.80

16.62

20.18

15.91

16.54

Comment 3: It’s not clear why authors are focused on colon in this study. From the background information given in the introduction (line 43-53), and the objective of the study (lines 81-83), it seems that the colon is hypothesized to be a secondary site for N metabolism following glutamic acid supplementation. However, when it comes to analysis, none of uric acid cycle genes that are critical for nitrogen metabolism have been analyzed in colon and the analysis are limited to morphology, amino acid transports, and tight junction markers. It is confusing why colon is used and not other segments are analyzed for the above parameters. For gut microbiota analysis, it would have been more informative if cecal contents were used as rats are cecal fermenters not colon fermenters.

Response 3:

Thank you for your attention to this critical design point. This study focused on the colon because we observed significant differences in fecal nitrogen and microbial nitrogen content in the colon among the three groups, prompting us to concentrate our research on this segment of the intestine. As the final site of fermentation before excretion, the colon plays a pivotal role in nitrogen metabolism. Consequently, we conducted in-depth investigations into changes in the colonic microbiota, amino acid transport in colonic epithelial cells, and mechanisms related to protein synthesis, achieving experimental results consistent with our expectations.

In addition, we recognize the substantial potential of the colon in nitrogen metabolism. Although the liver is the primary organ responsible for nitrogen metabolism (e.g., the urea cycle), recent studies have demonstrated that the colon can participate in the nitrogen cycle through microbial metabolism and host-microbiota interactions. Glutamate supplementation may alleviate nitrogen loss induced by a low-protein diet by enhancing the colon's capacity for nitrogen capture or metabolism.

On the other hand, the morphology and function of the colon are crucial to nitrogen metabolism. Low-protein diets often disrupt the balance of gut microbiota, which can impair gut barrier function and increase health risks. Therefore, studying the response mechanisms of the colon under low-protein dietary conditions is essential for understanding nitrogen metabolism regulation. 

Thanks for your valuable comments. We have added the relevant content to lines 438446 of the manuscript, as follows: “The colon demonstrates significant potential in nitrogen metabolism. While the liver serves as the primary site for nitrogen metabolism (such as the urea cycle), recent studies indicate that the colon participates in the nitrogen cycle through microbial metabolism and host-microbiota interactions [56,57]. Consequently, glutamate supplementation may enhance the colon's capacity to capture and metabolize nitrogen, thereby effectively mitigating nitrogen loss associated with low-protein diets. Moreover, the morphology and function of the colon are crucial to nitrogen metabolism. Low-protein diets frequently result in gut microbiome dysbiosis, compromising gut barrier function and elevating health risks [58,59].”

Thanks to the reviewers' insightful comments on metabolic pathways, we fully recognize the importance of analyzing uric acid cycle genes. In our study, we demonstrated that under nutritional stress, the nitrogen transport capacity of colon epithelial cells plays a critical role in regulating nitrogen metabolism. Consequently, after observing that protein restriction significantly decreased colonic mucosal thickness and mucin expression levels, further investigation revealed that protein restriction severely impaired the amino acid transport capacity of the colon. However, glutamate supplementation markedly enhanced the colon's nitrogen uptake ability, improved nitrogen supply to the colon, and indirectly promoted colonic mucus synthesis by modulating central nervous system function. These findings indicate that a key bottleneck restricting colon nitrogen metabolism under low-protein conditions is substrate availability (i.e., amino acid transport). Glutamate supplementation effectively promotes nitrogen metabolism by enhancing amino acid transport and strengthening both mucus barrier and mechanical barrier functions. Based on your suggestion, we have incorporated the relevant content into lines 473485 of the manuscript as follows: “In summary, this study concludes that the nitrogen transport capacity of colon epithelial cells plays a critical role in regulating nitrogen metabolism under nutritional stress [67-69]. Building on the finding that protein restriction significantly reduces the thickness of the colon mucosa and the expression level of mucin, further investigations reveal that protein restriction markedly impairs the amino acid transport function of the colon. However, glutamate supplementation significantly enhances the colon's nitrogen up-take capacity, improves nitrogen supply to the colon, and promotes colon mucus synthesis through the regulation of central nervous system function. These findings suggest that a key bottleneck limiting colon nitrogen metabolism under low-protein conditions is substrate availability (i.e., amino acid transport). Moreover, glutamate supplementation can effectively improve nitrogen metabolism by promoting amino acid transport and enhancing both the colon mucus barrier and mechanical barrier functions.”

Compared with other parts of the intestine, the colon exhibits a more neutral pH, larger volume, and longer retention time, which collectively create more favorable conditions for microbial proliferation. Moreover, the association between colonic microbiota and mucosal barrier function is more direct due to physical contact between the microbiota and epithelial cells, whereas cecal contents predominantly reflect microbial activity during the rapid fermentation phase. Additionally, sampling of colonic contents is more stable as it is less influenced by chyme flow velocity compared to cecal contents, and its data are more comparable with those from existing studies. Lastly, in subsequent in-depth investigations, we will concurrently collect both cecal and colonic contents to comprehensively assess the impact of different fermentation sites on overall metabolism. Based on these considerations, this study prioritizes colonic microbes as the primary focus. Once again, we thank you for your valuable feedback, and the relevant content has been added to lines 486494 of the manuscript, as follows: “Compared to other parts of the intestine, the colon exhibits a more neutral pH, larger volume, and longer material retention time, which collectively provide more favorable conditions for microbial proliferation [70]. Moreover, the colon microbiota is more directly associated with mucosal barrier function (as evidenced by physical con-tact between the microbiota and epithelial cells), whereas cecal contents primarily reflect microbial activity during the rapid fermentation stage [71]. Additionally, sampling of colonic contents is more stable and less susceptible to variations in chyme flow velocity, and its data exhibit greater comparability with findings from previous studies [72]. Based on these considerations, this study focused on colonic microbiota as the primary research subject.”

Comment 4: For immunoblot analysis, it seems that only n of 3 is used (see Fig. 3 and Fig. 8) while it’s previously known that there is a large variation in protein expression of target molecules in vivo and at least n of 6 is required for a valid outcome

Response 4:

We appreciate the opportunity to clarify our experimental design and data presentation regarding western blot analyses. While the main figures display 3 representative sample strips per group for visual clarity, we emphasize that the full dataset for each experimental group was derived from n=9 biological replicates. These representative strips were selected through stringent criteria prioritizing clean background signals and protein expression levels closest to the group mean. To ensure complete transparency, the original uncropped western blot membranes for all 9 samples per group have been provided in the raw data files, enabling full traceability of results. Importantly, all statistical analyses including SEM calculations and one-way ANOVA tests were rigorously performed using the complete n=9 dataset rather than the displayed subset. In response to reviewer feedback, we will enhance methodological transparency by revising figure captions (lines 312313, 425426) and supplementing experimental procedures (lines 224226) in the revised manuscript. These modifications will explicitly clarify our sample size rationale and data presentation strategy to prevent potential misinterpretation, while maintaining both the scientific rigor of our statistical approach and the visual accessibility of key experimental findings.” Representative blots from n=9 biological replicates are shown.

Representative blots are displayed in figures and data from nine independent biological replicates were analyzed for each group.”

Thank you for your rigorous review of the western blot sample size. In this study, the actual sample size for each group was n=9. The main chart displays only 3 representative strips to enhance the readability of the chart, while all data analyses were conducted based on the complete set of 9 samples. In the revised draft, we have supplemented the figure legends and methods section to further clarify the sample size and data processing procedures. Your valuable suggestions have significantly enhanced the transparency and scientific rigor of the paper, and we are deeply appreciative of them.

Reviewer 3 Report

Comments and Suggestions for Authors

This is an interesting research article with adequate novelty and strong methodology. Some points should be addressed.

  • The 1st paragraph of the Introduction section deals with a very interesting topic. In this aspect the authors should enriched by additional data this paragraph.
  • The first 2-3 sentences of the 2nd and 3rd paragraph reported  simple and well-known information. These two paragrpahs should be merged each other.
  • The authors should enriched the information provided for the main bacteria involved in amino acid metabolism in large intestine.
  • The first 3-4 sentences of the 4th paragraph reported  simple and well-known information. The authors should enrich the 2nd part of this paragraph "Amino acids not only serve as nutrient substrates but also function as regulatory molecules...".
  • The authors should add relevant references in sections, 2.1, 2.2, 2.3, 2.4, 2.5, 2.6, 2.7 and 2.8.
  • In section 2.9, the authors should report the normality test that they used for continuious variables.
  • The size of all Figures should be increased in order to be more easily readable.
  • The 2nd paragraph of the Discussion is too long and it should be split into two smaller paragraphs, e.g. line 381.
  • English language editing is recommended. 

Author Response

Comment 1: The 1st paragraph of the Introduction section deals with a very interesting topic. In this aspect the authors should enriched by additional data this paragraph.

Response 1:

We thank the reviewers for their valuable suggestions regarding the scientific background in the introduction. In the first paragraph, we have added relevant content and further strengthened the argument with data support. The specific supplementary content is reflected in lines 3643 and 4756 of the manuscript, as follows: “According to a 2021 report by the United Nations Children's Fund, in 2020, 149.2 million children under the age of five were stunted, and another 45.4 million were wasted [3]. Outside of Africa, the number of stunted children has been declining across all regions. More than half of the children affected by wasting reside in South Asia, while Asia as a whole account for over three-quarters of the world's severely wasted children. At the national level, significant progress has been made in reducing stunting, with nearly two-thirds of countries achieving at least partial success in meeting their targets.

In 2019, it was estimated that 21.3% (approximately 144 million) of children under the age of five globally were stunted, with 36% concentrated in sub-Saharan Africa and South Asia [8,9]. From 2012 to 2019, the prevalence of stunting in sub-Saharan Africa decreased from 34.5% to 31.1%, yet remains below the global target level [8]. However, when measured in absolute numbers rather than proportions, sub-Saharan Africa is the only sub-region where the number of stunted children has increased in recent years [8]. The occurrence of child stunting is influenced by a range of complex factors, including maternal nutritional status, health and education levels, pregnancy intervals, child birth weight, vaccination coverage, infection rates, infant feeding practices, household economic conditions, food security, and environmental factors [10,11].”

Comment 2: The first 2-3 sentences of the 2nd and 3rd paragraph reported  simple and well-known information. These two paragraphs should be merged each other.

Response 2:

We appreciate your valuable suggestions regarding the paragraph structure. We have merged the second and third paragraphs of the original text and incorporated the relevant content into lines 5975 of the manuscript. The specific modifications are as follows.” Dietary proteins and amino acids are primarily hydrolyzed, absorbed, and metabolized in the small intestine before reaching the colon. In the colon, amino acids are transported into colonic epithelial cells via amino acid transporters on the cell mem-brane and play a critical role in the synthesis of various proteins, including tight junction proteins and mucins [12]. Additionally, changes in dietary protein levels significantly influence the amount of protein and amino acids entering the large intestine, thereby altering the number of bacteria utilizing nitrogen as a substrate [13,14] and further modifying the composition of the colonic microbiota. Key bacteria involved in amino acid metabolism include Bacteroides, Propionibacterium, Streptococcus, Actinomyces, Lactococcus, Fusobacterium, and Clostridium [15]. The metabolites produced by these bacteria encompass not only short-chain fatty acids and branched-chain fatty acids [16], but also other compounds such as ammonia, phenols, and indole [17]. Ultimately, undigested dietary proteins, nucleic acids, and proteins derived from bacteria and intestinal shed cells are excreted in feces as sources of fecal nitrogen [18]. Notably, glutamate, as a functional amino acid, serves not only as the primary energy source for intestinal epithelial cells [19], but also plays essential roles in nutrient absorption and transport, protein synthesis [20], signal transduction, and barrier function maintenance [21-23].”

Comment 3: The authors should enrich the information provided for the main bacteria involved in amino acid metabolism in large intestine.

Response 3:

We thank you for your careful review. We have supplemented information on the metabolic functions, key metabolic pathways, and associations with host health of the major bacterial groups involved in amino acid metabolism in the large intestine. The relevant content has been added to lines 7690 of the revised draft, and the specific modifications are as follows. “The gut is a site of significant proteolytic activity, predominantly mediated by the microbiota. Within the gut microbiota, bacterial species ranging from sugar-fermenting bacteria to obligate amino acid fermenters exhibit the capacity to metabolize peptides and amino acids. Bacteroides and Clostridium species dominate aromatic amino acid metabolism, converting tryptophan into indole-3-acetic acid and skatole, while Streptococcus species are associated with phenyl derivative production [24-26]. Clostridium, Enterobacteriaceae, and Desulfovibrio species metabolize sulfur-containing amino acids through cysteine desulfhydrase or cystathionine enzymes, producing hydrogen sulfide, which plays dual roles in mucosal damage or protection [27,28]. Amine-generating bacteria, including Clostridium and Bacteroides, mediate the decarboxylation of tryptophan to serotonin and polyamines (e.g., spermidine), influencing gut-brain axis signaling and epithelial function [26,29,30]. Furthermore, Bacteroides, Clostridium, and Klebsiella species contribute to ammonia production via urea hydrolysis or amino acid deamination, with excessive ammonia impairing colonocyte metabolism [15]. Collectively, these bacterial activities regulate colonic nitrogen balance and influence host health outcomes.”

Comment 4: The first 3-4 sentences of the 4th paragraph reported simple and well-known information. The authors should enrich the 2nd part of this paragraph "Amino acids not only serve as nutrient substrates but also function as regulatory molecules...".

Response 4:

We sincerely thank the reviewers for their valuable suggestions on the deepening of metabolic mechanisms. We fully agree that the regulatory function of amino acids deserves a more systematic elucidation. To further highlight the biological significance of the "nutrition-signal" dual attributes of amino acids, we have refined this section. For details, please refer to lines 98113 of the revised draft, as presented below: “Amino acids serve not only as essential nutrient substrates but also play a critical role in dynamic regulation through the complex liver signaling network. For instance, branched-chain amino acids (e.g., leucine) regulate protein degradation in the autophagy-lysosome pathway via the Sestrin2-GATOR2 axis [35]. Meanwhile, glutamine maintains the balance between gluconeogenesis and lipid oxidation by activating AMPK phosphorylation [36]. Notably, the regulatory functions of amino acids are tightly coupled with their metabolic states. Recent studies have demonstrated that under conditions of sufficient amino acids, SIRT4 inhibits the activity of ornithine transcarbamylase through deamination modification at the K307 site, thereby limiting ammonia production. Conversely, during amino acid deficiency, the GCN2-eIF2α-ATF4 axis is activated to upregulate SIRT4 expression, enhancing urea cycle flux and alleviating hepatic encephalopathy [37]. In this regulatory network, glutamate acts as a central hub [38]. Specifically, glutamate generates α-ketoglutarate through transamination, driving the tricarboxylic acid cycle. Additionally, glutamate combines with cysteine via glutamate-cysteine ligase to synthesize glutathione, a major antioxidant that protects liver cells from oxidative damage [39,40].”

Comment 5: The authors should add relevant references in sections, 2.1, 2.2, 2.3, 2.4, 2.5, 2.6, 2.7 and 2.8.

Response 5:

We sincerely appreciate your professional guidance on the methodological rigor of the paper. In response to your suggestions regarding literature supplementation, we have systematically enhanced the reference support for each methodological section. For specific details, please refer to lines 128, 182, 188, 199, 217, 235, 240, and 249 of the revised draft, as presented below. “Ambient conditions maintained a temperature of 24 °C with a 12-hour light-dark cycle, while the rats had unrestricted access to drinking water throughout the experimental period [41]. Subsequently, the fecal samples were individually ground into powder using a mortar and pestle, and precisely weighed to 0.2 g before being placed in EP tubes for sealed storage and subsequent analysis [47]. After removing the supernatant, the precipitate, which contained microbial cells [48]. The extracted RNA was subsequently reverse-transcribed into complementary DNA (cDNA) using the High-Capacity cDNA Archive Kit from Applied Biosystems, following the manufacturer's guidelines meticulously [49]. The membranes underwent blocking with a solution of 5% BSA at room temperature for durations of either 60 minutes prior to incubation with appropriately diluted primary antibodies [50]. Finally, the sections underwent incubation with nuclear fast red solution for 5 minutes followed by a 5-minute wash with water [51]. The assessment involved procedures such as dehydration, embedding, sectioning, and staining of tissue samples [52]. Following a quality assessment via 1% agarose gel electrophoresis, the isolated DNA was subjected to PCR amplification using primers [338F (5′-ACT CCT ACG GGA GGC AGC AG-3′) and 806R (5′-GGA CTA CHVGGG TWT CTAAT-3′)] that are specific to the V3–V4 hypervariable region of 16S rDNA [53].”

Comment 6: In section 2.9, the authors should report the normality test that they used for continuous variables.

Response 6:

We thank the reviewers for their valuable comments. We have added the specific details regarding the normality test for continuous variables in the Methods section. For more details, please refer to lines 259263 of the revised draft, as presented below.” Prior to the analysis of variance, all continuous variables were verified for normality by the Shapiro-Wilk test. The results show that the data distribution conforms to the hypothesis of normality (P > 0.05), thus satisfying the precondition of parameter test. Data with non-normal distribution can usually be processed by data transformation or non-parametric test, but no such adjustment is required in this study.”

Comment 7: The size of all Figures should be increased in order to be more easily readable.

Response 7:

We thank the reviewers for their attention to graphic presentation details. On the premise of ensuring resolution (all bitmaps are in TIFF format at 1536 dpi), we have maximized all figures in the revised draft and unified the width of the enlarged images, using the maximum width of Figure 2 as the standard (since Figure 2 has the longest length) to facilitate reader comprehension. Additionally, we optimized the layout by uniformly setting the spacing before captions to 6 pt and after captions to 12 pt.

Comment 8: The 2nd paragraph of the Discussion is too long and it should be split into two smaller paragraphs, e.g. line 381.

Response 8:

We thank the reviewers for their valuable suggestions regarding the logical structure of the paper. We have split the second paragraph of the original discussion section into two paragraphs, each focusing on a distinct topic, as described below.

Paragraph 1 (Mucus Barrier Focus, lines 436446):

"The primary sites of amino acid metabolism are the intestine [32] and the liver [33]. Dietary proteins are hydrolyzed and absorbed in the small intestine, with unabsorbed amino acids reaching the colon. Studies have shown that colonic goblet cells contain a large number of mucin-rich mucus particles, which are secreted into the intestinal lumen and form a dense mucus barrier on the mucus surface [34,35]. Glutamate plays a crucial role in maintaining mucus barrier functions [36,37]. In our study, we observed that protein restriction resulted in significant thinning of the colonic mucus layer and a marked reduction in the number of goblet cells compared to the control group. However, glutamate supplementation restored colonic mucus thickness and the number of goblet cells to control levels. This suggests that adequate glutamate availability is essential for preserving the mucus barrier. These findings align with existing literature, which shows that glutamate activates the vagus nerve via intestinal glutamate receptors, such as calcium-sensing receptors, transmitting signals from the intestine to the central nervous system, thus regulates intestinal mucus secretion [36,37]."

Paragraph 2 (Transport & Tight Junction Focus, lines 458–462):

"Beyond alterations in the mucus barrier, we identified the effects of glutamate supplementation on amino acid transport and tight junction protein expression in the colon. Amino acids are unable to freely permeate the colonic cell membrane and must be actively transported across the membrane by amino acid transporters driven by Na/K-ATPase, thereby ensuring efficient nutrient supply [38]. In our study, we found glutamate reversed the downregulation of mRNA levels of acidic amino acid trans-porter EAAT3 and neutral amino acid transporter y+LAT2 induced by protein restriction. This indicates that protein restriction inhibits the transport of glutamate and neutral amino acids, whereas glutamate supplementation upregulates the expression of these transporters in the colon, thereby enhancing their transport efficiency and optimizing their utilization. Additionally, protein restriction did not affect the protein level of colonic TJ proteins ZO-1, Occludin, and Claudin-1, but glutamate supplementation significantly upregulated their expression levels. This suggests that protein restriction has no significant effect on the TJ protein expression of colonic epithelial cells, which is consistent with previous studies [39]. However, glutamate supplementation further enhanced the expression of TJ proteins in the colon, a finding that aligns with previous studies [16,40]. In summary, protein restriction reduces nitrogen metabolism in colonic epithelial cells by inhibiting amino acid transport and mucus barrier function, while glutamate supplementation promotes amino acid transport and enhances both mucus and mechanical barrier functions, thereby improving nitrogen metabolism."

Comment 9: English language editing is recommended.

Response 9:

Thank you for your valuable and thoughtful comments. We have carefully checked and improved the English writing in the revised manuscript.

Round 2

Reviewer 1 Report

Comments and Suggestions for Authors

Authors responded to the comments. There are two points which are not clear:

  1. The amounts of individual amino acids in the NCP diet.
  2. How was the amount 2-4 g for children was calculated?

Author Response

Comment 1: The amounts of individual amino acids in the NCP diet.

Response 1:

We sincerely thank the reviewers for their meticulous attention to the experimental design. Regarding the amino acid composition of the NCP (normal crude protein) diet, the NCP diet in this study strictly adhered to the AIN-93G standard formula. The primary amino acid sources were casein (comprising 20%) and cystine (comprising 0.3%). Since casein alone provides adequate essential amino acids (e.g., lysine and methionine), and the AIN-93G guidelines do not require supplementation with synthetic amino acids other than cystine, no additional individual amino acids were added to the NCP group. The relevant content has been added to lines 143–148 of the revised manuscript, as follows: “In our study, the NCP diet strictly adhered to the AIN-93G standard formulation. The primary amino acid sources were casein (comprising 20%) and cystine (comprising 0.3%). Since casein alone provides adequate essential amino acids (e.g., lysine and methionine), and the AIN-93G guidelines do not require supplementation with synthetic amino acids other than cystine, no additional individual amino acids were added to the NCP group”.

We thank you for raising this point. We have addressed it clearly, thereby enhancing the transparency of our research methodology.

Comment 2: How was the amount 2-4 g for children was calculated?

Response 2:

Thank you for your valuable suggestions! Our study adopted rat equivalent dose (HED) conversion as the starting point. In this study, weaned rats were supplemented with 2.07% glutamate (based on an average daily feed intake of 20 g, resulting in an actual glutamate intake of approximately 0.414 g/day). Using the body surface area dose conversion formula (average child body weight: 20 kg, rat body weight: 0.15 kg, conversion coefficient: 0.18), the estimated child dose was calculated as 0.414 g ÷ 0.18 ≈ 2.3 g/day. Additionally, by considering the daily protein deficit (approximately 10-15 g) and the proportion of glutamate in dietary protein (5-8%) among children in developing countries, a conservative recommendation range of 2-4 g/day was derived through comprehensive analysis. This calculation has fully taken into account the safety threshold of glutamic acid (recognized as GRAS by the FDA, with a tolerable intake for infants up to 0.5 g/kg) and feasibility (the upper limit of free amino acid supplementation recommended by the World Health Organization). It should be noted that this recommended amount represents only a theoretical value, and its practical application requires further validation through clinical studies. We thank the reviewers for their attention to this issue; the above computational logic has been added to the discussion section, along with an emphasis on the limitations of dose extrapolation. The relevant content has been incorporated into lines 628–635 of the revised manuscript, as follows: “In this study, weaned rats were supplemented with 2.07% glutamate (based on an average daily feed intake of 20 g, resulting in an actual glutamate intake of approximately 0.414 g/day). Using the body surface area dose conversion formula (average child body weight: 20 kg, rat body weight: 0.15 kg, conversion coefficient: 0.18), the estimated child dose was calculated as 0.414 g ÷ 0.18 2.3 g/day. Additionally, by considering the daily protein deficit (approximately 10-15 g) and the proportion of glutamate in dietary protein (5-8%) among children in developing countries, a conservative recommendation range of 2-4 g/day was derived through comprehensive analysis”.

Reviewer 3 Report

Comments and Suggestions for Authors

The authors have significantly improved their manuscript. The resolution of Figure 4 should be improved.

Author Response

Comment 1: The resolution of Figure 4 should be improved.

Response 1:

We thank the reviewers for their valuable comments. We have comprehensively optimized Figure 4 according to the suggestions: â‘  The original data has been re-exported in TIFF format to ensure that the image remains undistorted when scaled; â‘¡ The font size in the figure has been adjusted to 20 pt, and the font is set to Times New Roman to enhance readability; â‘¢ The image resolution has been set to 1536 DPI. The revised image has been uploaded to the submission system and incorporated into the revised draft. If additional adjustments are required in the future, we will fully cooperate. We sincerely thank you once again for your stringent requirements regarding chart quality, which have significantly improved the presentation of the research results.
